# EMBO
*reports*

# Adducin-1 is essential for spindle pole integrity through its interaction with TPX2

Wen-Hsin Hsu[1], Won-Jing Wang[2], Wan-Yi Lin[3], Yu-Min Huang[4], Chien-Chen Lai[4], Jung-Chi Liao[5] & Hong-Chen Chen[1,2,3,6,7,*]

## Abstract

Bipolar spindle assembly is necessary to ensure the proper progression of cell division. Loss of spindle pole integrity leads to multipolar spindles and aberrant chromosomal segregation. However, the mechanism underlying the maintenance of spindle pole integrity remains unclear. In this study, we show that the actin-binding protein adducin-1 (ADD1) is phosphorylated at S726 during mitosis. S726-phosphorylated ADD1 localizes to centrosomes, wherein it organizes into a rosette-like structure at the pericentriolar material. ADD1 depletion causes centriole splitting and therefore results in multipolar spindles during mitosis, which can be restored by re-expression of ADD1 and the phosphomimetic S726D mutant but not by the S726A mutant. Moreover, the phosphorylation of ADD1 at S726 is crucial for its interaction with TPX2, which is essential for spindle pole integrity. Together, our findings unveil a novel function of ADD1 in maintaining spindle pole integrity through its interaction with TPX2.

**Keywords** adducin; centrosome; mitosis; spindle; TPX2
**Subject Categories** Cell Adhesion, Polarity & Cytoskeleton; Cell Cycle; Post-translational Modifications, Proteolysis & Proteomics

## Introduction

To achieve faithful chromosome segregation, a bipolar spindle has to be established. Correct reproduction and structural organization of centrosomes are crucial for the establishment of spindle bipolarity [1,2]. Multipolar spindles are often seen in human tumors and are usually associated with multiple centrosomes and chromosomal instability [3]. Centrosome overduplication and cytokinesis failure lead to multiple centrosomes and multipolar spindles [4]. In addition, multipolar spindles can also result from the loss of spindle pole integrity. Each bipolar spindle pole or centrosome consists of engaged mother and daughter centrioles surrounded by organized pericentriolar materials (PCMs) [5]. The loss of spindle pole integrity can lead to PCM fragmentation or centriole splitting, resulting in excess centrosomes and multipolar spindles during mitosis [6].

The inefficient inhibition of separase, a cysteine protease, during mitotic delay or arrest may result in both unscheduled sister chromatid separation and premature centriole disengagement, followed by multipolar spindle formation [7,8]. Pericentrin (PCNT), a PCM protein that protects engaged centrioles from premature disengagement, was identified as a crucial substrate for separase at centrosomes [9,10]. In addition, astrin and Aki1 have been shown to regulate centrosomal separase activity. Depletion of either astrin or Aki1 induces centriole splitting and multipolar spindles [11,12]. Increasing evidence indicates that the balance of motor-driven forces acting on centrosomes helps hold mother and daughter centrioles during mitosis [6,13]. The antagonistic function of the motor proteins Eg5 (also known as kinesin-5 and KIF11) and dynein within centrosomes is important for spindle pole integrity [14].

TPX2 is a conserved multifunctional microtubule-associated protein that is required for spindle assembly and function. It localizes in the nucleus during interphase and associates with mitotic spindles during mitosis [15,16]. In addition to spindle fibers, TPX2 localizes to spindle poles and is required for spindle pole integrity [16–19]. TPX2 contributes to microtubule polymerization for mitotic spindle assembly in a Ran-dependent manner [20,21]. The interaction between TPX2 and myosin-X (Myo10), an unconventional myosin, is crucial for spindle assembly [22]. TPX2 binds to the mitotic kinase Aurora-A through its NH$_2$-terminus, which is essential for spindle localization and activation of Aurora-A [23,24]. The interaction between TPX2 and Aurora-A is essential for the control of spindle length [17]. In addition, TPX2 binds Eg5 through its COOH-terminus, which localizes Eg5 to spindle fibers and regulates its motor behavior [25]. TPX2 was shown to slow the motion of

1   Ph.D. Program in Tissue Engineering and Regenerative Medicine, National Chung Hsing University, Taichung, Taiwan
2   Institute of Biochemistry and Molecular Biology, National Yang-Ming University, Taipei, Taiwan
3   Institute of Biomedical Sciences, National Chung Hsing University, Taichung, Taiwan
4   Institute of Molecular Biology, National Chung Hsing University, Taichung, Taiwan
5   Institute of Atomic and Molecular Sciences, Academia Sinica, Taipei, Taiwan
6   Department of Life Sciences, National Chung Hsing University, Taichung, Taiwan
7   Cancer Progression Research Center, National Yang-Ming University, Taipei, Taiwan
    *Corresponding author. Tel: +886 2 28267123; Fax: +886 2 28201886; E-mail: hcchen1029@ym.edu.tw

Eg5, which plays a key role in regulating the forces required for spindle bipolarity [26]. More recently, TPX2 was shown to be a centrosomal protein and play an important role in Eg5-dependent centrosome separation before nuclear envelope breakdown [27].

Adducin (ADD) is a membrane actin-binding protein that is mainly localized at actin–spectrin junctions [28,29]. The adducin family comprises three closely related genes (α, β, and γ). The α (ADD1) and γ (ADD3) forms are found in most tissues, whereas the β form (ADD2) is abundant in the brain and erythrocytes [30]. All adducin proteins are similar in their amino acid sequences and domain structures, which consist of an amino-terminal head domain, a neck domain, and a carboxyl-terminal protease-sensitive tail domain [31,32]. The tail domain is composed of a carboxyl-terminal stretch of 22 residues with homology to the myristoylated alanine-rich C kinase substrate (MARCKS) domain [29,32]. Adducin interacts with spectrin, calmodulin, and F-actin through the MARCKS-related motif [29,32,33]. Phosphorylation in this motif by protein kinase C or protein kinase A decreases its ability to bind spectrin and F-actin [34,35]. Adducin has been shown to be important for erythrocyte membrane skeleton assembly [30], cell–cell junction stabilization [36,37], neuronal synapse assembly [38], axon diameter maintenance [39], and cell migration facilitation [40,41]. Recently, ADD1 was shown to play an important role in mitosis [42]. The phosphorylation of ADD1 at residues S12 and S355 by cyclin-dependent kinase 1 (CDK1) allows ADD1 to interact with Myo10 and localize to mitotic spindles, which is essential for proper spindle assembly [42]. In this study, we show that ADD1 is phosphorylated at residue S726 and localized to spindle poles during mitosis. ADD1 phosphorylation at S726 is important for its interaction with TPX2, which is crucial for spindle pole integrity.

## Results

### S726-phosphorylated ADD1 localizes at mitotic centrosomes and organizes into a rosette-like structure at PCMs

Our mass spectrometry analysis revealed that S726 is a phosphorylation site on ADD1 in mitotic cells (Appendix Fig S1). Indeed, the phosphorylation level of ADD1 at S726 was increased in the M phase, but it returned to the baseline upon cell cycle progression to the G1 phase, as analyzed by immunoblotting with an antibody specific to S726-phosphorylated ADD1 (Fig 1A). Surprisingly, S726-phosphorylated ADD1 localized at mitotic centrosomes (Fig 1B). The specificity of ADD1 pS726 staining at mitotic centrosomes was

verified by ADD1 depletion and a phosphopeptide (pS726 peptide) corresponding to the residues surrounding ADD1 S726, which is a competitor for the phospho-specific antibody (Figs 1C and EV1). S726-phosphorylated ADD1 was hardly detected at the G1- and S-phase centrosomes but visible at the G2-phase centrosomes (Fig 1D and Appendix Fig S2). During prophase and metaphase, S726-phosphorylated ADD1 was more apparently localized at the centrosomes and organized into a rosette-like structure at the PCM (Fig 1D and Appendix Fig S2). During late mitosis (telophase), S726-phosphorylated ADD1 was no longer detected at the centrosomes (Fig 1D and Appendix Fig S2). Using SAS6 as a marker for daughter centrioles and centrin1 as a marker for the centriole distal ends, we found S726-phosphorylated ADD1 to be closely localized to the proximal ends of mother centrioles (Fig 1E).

Super-resolution structured illumination microscopy (SIM) revealed that S726-phosphorylated ADD1 is a PCM component that forms a rosette-like structure inside the "beads-on-string" structure formed by PCNT (Fig 2A). In interphase centrosomes, S726-phosphorylated ADD1 was closely distributed with γ-tubulin and NEDD1 (Fig 2B and C). During mitosis, additional, less organized PCM proteins were recruited to the centrosomes, wherein S726-phosphorylated ADD1 tends to form a rosette-like structure, and it mingled with PCNT, γ-tubulin, and NEDD1 (Fig 2D–F). Direct stochastic optical reconstruction microscopy (dSTORM) confirmed that the rosette-like structure formed by S726-phosphorylated ADD1 in the M phase was larger ($0.3 \pm 0.05$ μm in diameter) than that in the G2 phase (Fig 2G), indicating that the rosette-like structure formed by S726-phosphorylated ADD1 is expanded during mitotic PCM expansion.

### ADD1 phosphorylation at S726 is important for spindle pole integrity

As shown previously [42], ADD1 depletion led to disorganized mitotic spindles, which was characterized by distorted spindles, elongated spindle length, and multipolar spindles (Fig EV2). The ADD1 S726A mutant was able to restore the distortion and elongation defects of mitotic spindles but not the multipolar spindle defect (Fig EV2), suggesting that S726 phosphorylation may be important for the function of ADD1 at spindle poles rather than spindle fibers. We further demonstrated that the multipolar spindle defect resulted from multiple spindle poles (as manifested by multiple γ-tubulin foci), which were restored by FLAG-ADD1 and the phosphomimetic S726D mutant but not by the S726A mutant (Fig 3A–C). The multiple spindle poles caused by ADD1 depletion did not result from

**Figure 1. S726-phosphorylated ADD1 is localized at mitotic centrosomes.**

A  HeLa cells remained asynchronized (Async.) or were synchronized in the M phase. The M phase-arrested cells proceeded to the G1 phase after nocodazole was removed for 4 h. Whole-cell lysates were analyzed by immunoblotting (IB) with the indicated antibodies.
B  HeLa cells were stained for ADD1 pS726, α-tubulin, and DNA. High magnification images of the cells in the white box are shown on the right. Scale bar, 5 μm.
C  The specificity of the immunofluorescence staining with anti-ADD1 pS726 was verified by ADD1 depletion (sh-ADD1) and competition with the pS726 peptide. Scale bars, 5 μm.
D  RPE1 cells at the indicated cell cycle phases were stained for ADD1 pS726, centrin1, γ-tubulin, and DNA. High magnification images of each centrosome are shown on the right. Centrin1 is a marker for centrioles. Scale bars, 10 μm (main image) and 1 μm (zoomed images).
E  RPE1 cells at the G2 and metaphases were stained for ADD1 pS726, SAS6, centrin1, and DNA. SAS6 is a marker for daughter centrioles. Scale bars, 10 μm (main image) and 1 μm (zoomed images).

Source data are available online for this figure.

centrosome overduplication or centriole disengagement during interphase (Fig EV3). Instead, we found that most abnormal spindle poles caused by ADD1 depletion contained only one centriole (Fig 3D and E), indicating that ADD1 depletion causes centriole splitting in mitosis and therefore results in multiple spindle poles. It has been known that inhibition of Aurora-B abolishes cytokinesis and leads to chromosomal instability [43]. Upon entry into mitosis, the Aurora-B-deficient cells contain extra numbers of centrosomes

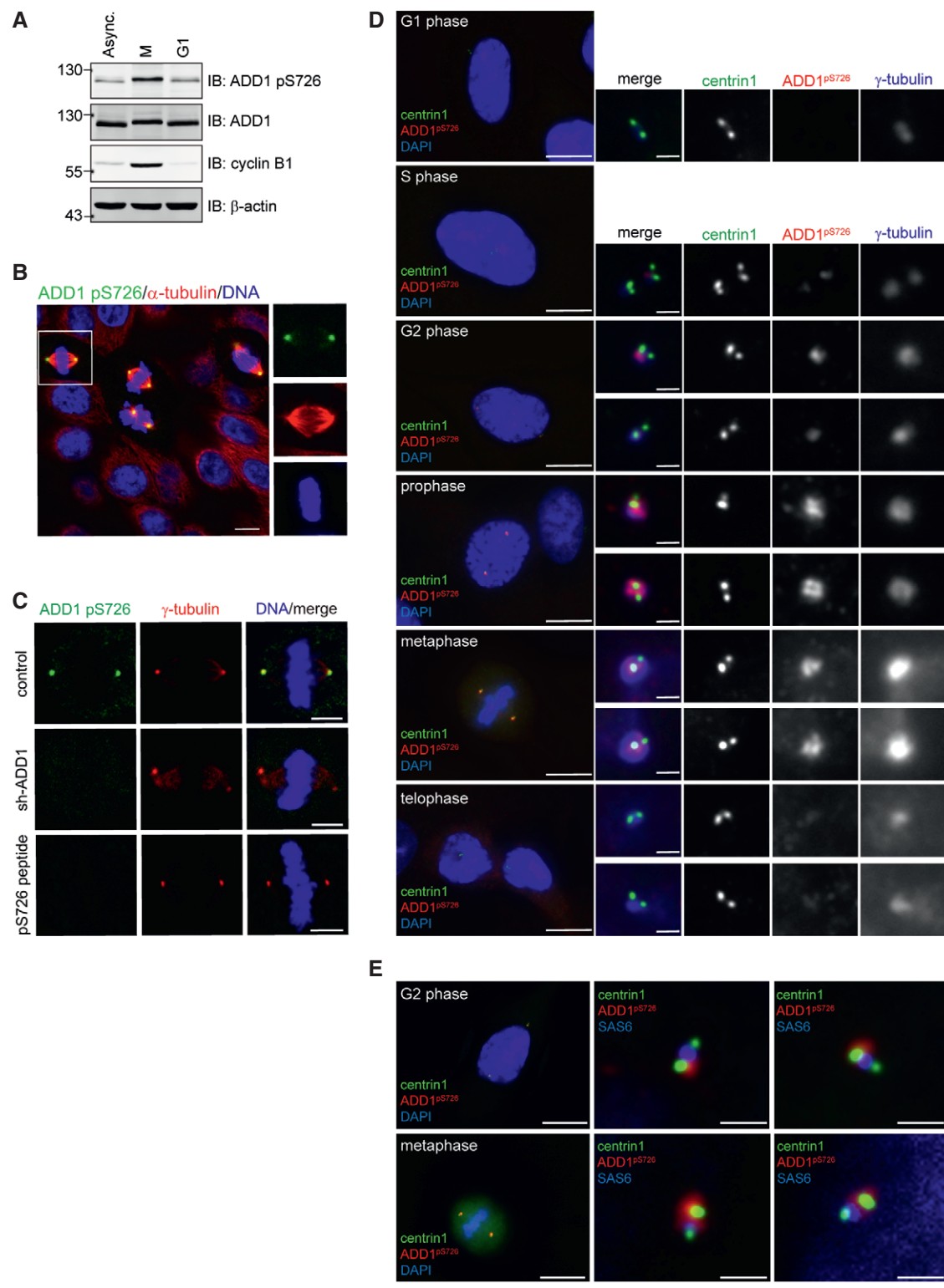

**Figure 1.**

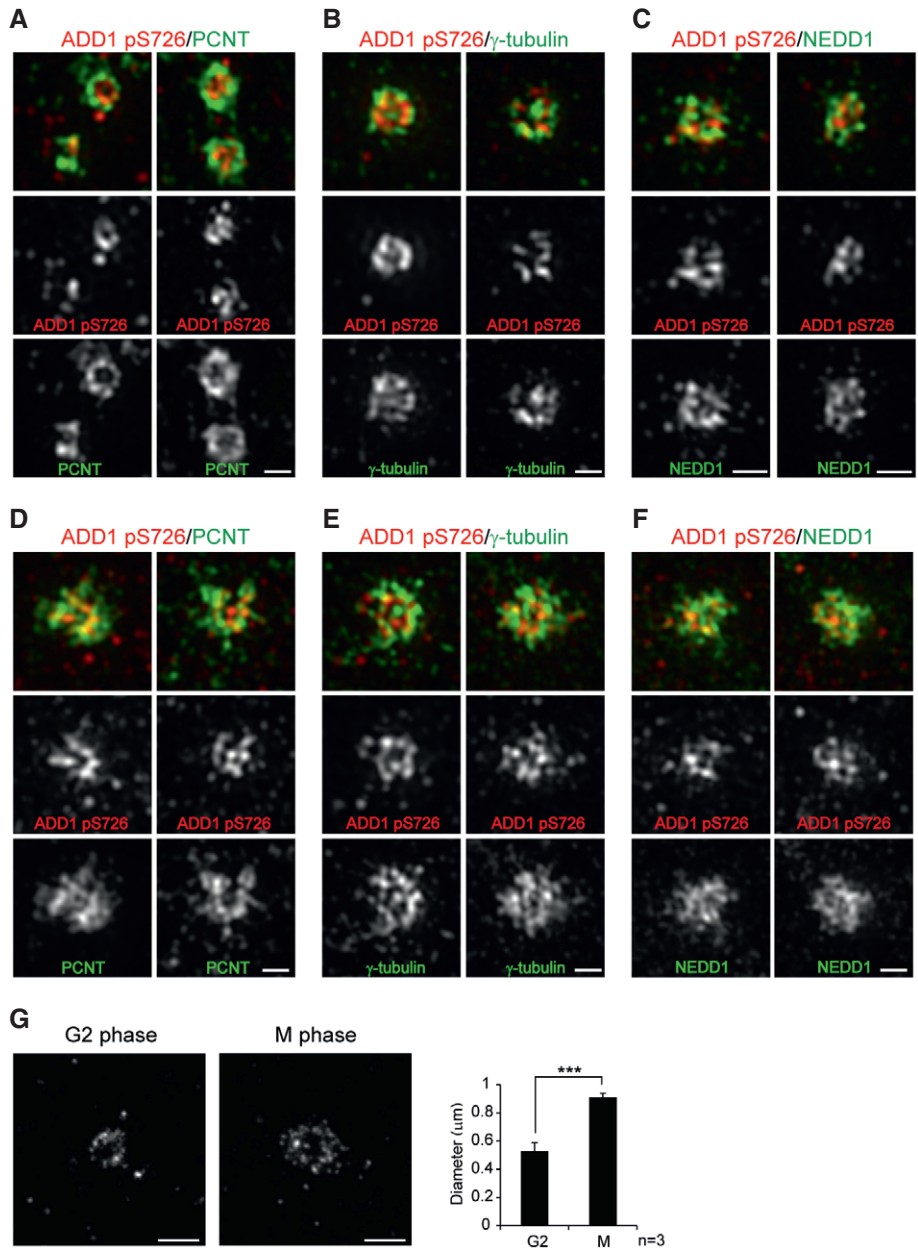

**Figure 2. S726-phosphorylated ADD1 is organized into a rosette-like structure at the PCM.**

A–F    PRE1 cells at the interphase (A–C) or M phase (D–F) were stained for ADD1 pS726, PCNT, γ-tubulin, and NEDD1. The images were acquired by SIM super-resolution microscopy. Scale bars, 0.5 μm.

G    RPE1 cells at the G2 or M phases were stained with an anti-ADD1 pS726 antibody, and the images were acquired using dSTORM super-resolution microscopy. Scale bars, 0.5 μm. The maximum diameter of the ADD1 pS726 staining signal was measured. Data are presented as means ± s.d. ***P = 0.0006 (Student's t-test).

and form multipolar spindles with intact spindle pole integrity. In the control experiment, the multiple spindle poles caused by Aurora-B depletion contained two centrioles (Fig 3D and E). Since ~ 50% of the multiple spindle poles occurred within 1 h upon entry into mitosis (Fig EV4), the centriole splitting caused by ADD1 depletion cannot be solely attributed to a mitotic arrest. Together, these results suggest that ADD1 phosphorylation at S726 may be important for spindle pole integrity. Premature separase activation leads to centriole splitting during mitosis [11,12]. However, depletion of

ADD1 does not lead to premature separase activation during mitosis (Appendix Fig S3).

## ADD1 phosphorylation at S726 is crucial for its interaction with TPX2

TPX2 is a microtubule-associated protein that plays important roles in spindle assembly and centrosome integrity [16–19]. Given that Myo10 interacts with TPX2 [22] and ADD1 [42], this prompted us to

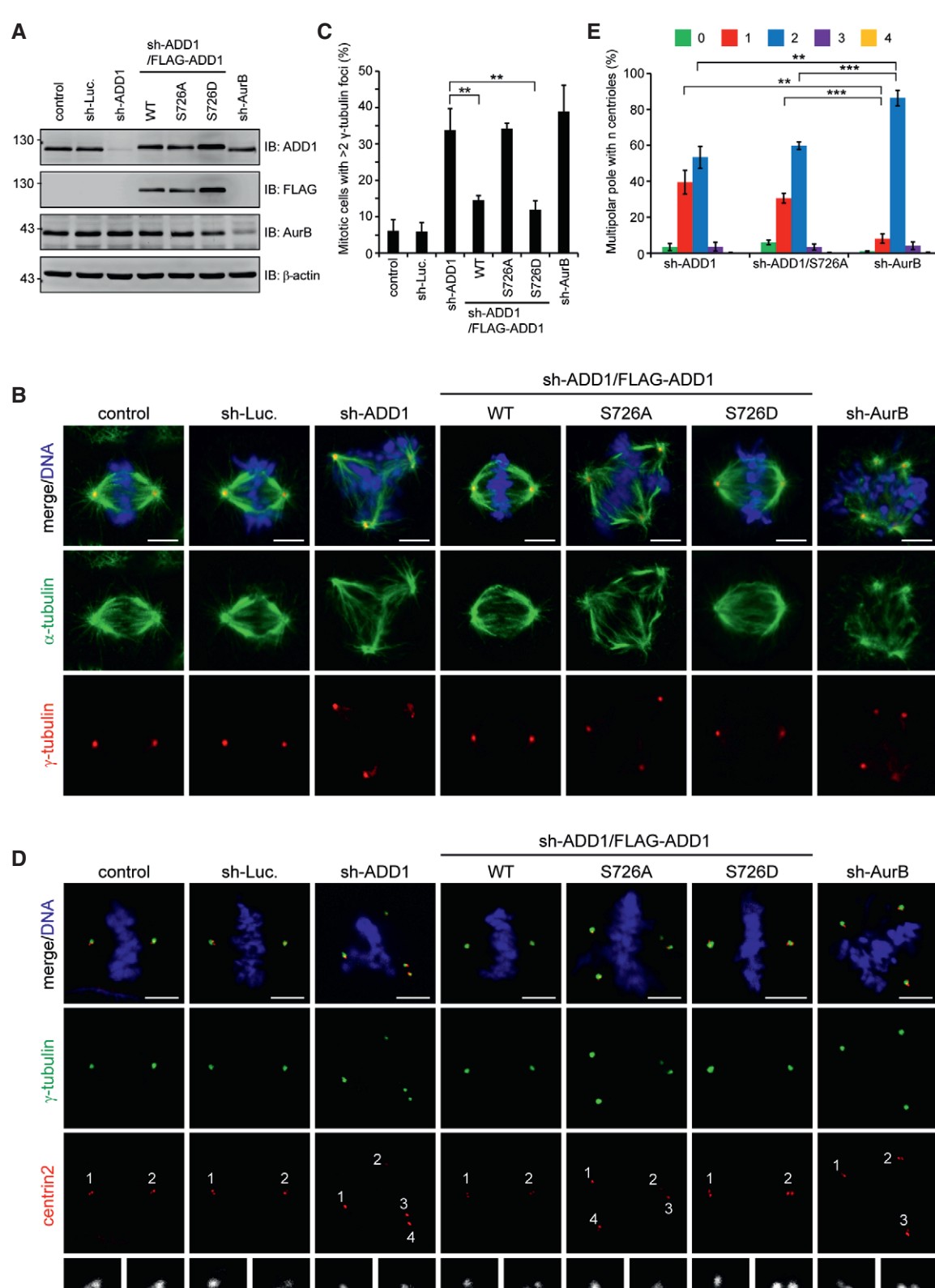

Figure 3.

**Figure 3. Supernumerary centrosomes induced by ADD1 depletion result from centriole splitting during mitosis.**

A   HeLa cells were infected with lentiviruses expressing shRNAs specific to ADD1 (sh-ADD1), Aurora-B (sh-AurB), or luciferase (sh-Luc.). FLAG-ADD1 WT or mutants (S726A and S726D) were re-expressed into the cells whose endogenous ADD1 had been depleted. Equal amounts of whole-cell lysates were analyzed by immunoblotting (IB) with the indicated antibodies.

B   The cells were stained for γ-tubulin (red), α-tubulin (green), and DNA (blue). Scale bars, 5 μm.

C   The percentage of multiple γ-tubulin foci in the total counted mitotic cells was measured (302–568 mitotic cells were counted in each group).

D   The cells were stained for γ-tubulin (green), centrin2 (red), and DNA (blue). The insets show magnification of the centrin2 signal at the indicated poles. Scale bars, 5 μm.

E   The percentage of spindle poles with *n* centrioles found in cells with multiple γ-tubulin foci was assessed (411–805 poles were counted in each group).

Data information: In (C and E), values (means ± s.d.) are from three independent experiments. \*\*$P < 0.01$ and \*\*\*$P < 0.001$ (Student's *t*-test). For clarity, not all significances are indicated.

Source data are available online for this figure.

---

examine whether TPX2 interacts with ADD1 through Myo10. Indeed, we found that TPX2 interacted with ADD1 in mitosis, but the interaction was not detected in interphase (Fig 4A). However, we cannot demonstrate a ternary complex containing TPX2, ADD1, and Myo10; instead, our results support a direct interaction between ADD1 and TPX2 (see below). Substitution of S726 with Ala significantly (~ 50%) decreased the interaction of ADD1 with TPX2 (Fig 4A), suggesting that S726 phosphorylation may be important for ADD1 to interact with TPX2. Moreover, S726-phosphorylated ADD1 and TPX2 were detected in the mitotic centrosome-enriched fractions (Fig 4B) and co-localized at the spindle poles (Fig 4C). While TPX2 was not detected in the interphase centrosomes, it apparently localized at the centrosomes during prophase and metaphase (Fig 4D). To identify the minimal region of TPX2 sufficient for ADD1 binding, a series of TPX2 constructs were generated and subjected to *in vitro* pull-down assays. The ADD1-binding region of TPX2 was located within aa 120–370 (Fig 4E), which was sufficient to bind purified FLAG-ADD1 *in vitro* (Fig 4F), supporting a direct interaction between TPX2 and ADD1. The TPX2 aa 120–370 fragment was able to bind the tail domain of ADD1 (Fig 4G), but less sufficient in binding the S726A mutant *in vitro* (Fig 4H), suggesting that ADD1 S726 phosphorylation is crucial for its interaction with TPX2.

**The interaction of TPX2 with both ADD1 and Eg5 is important for spindle pole integrity**

To dissect the significance of Aurora-A, ADD1, and Eg5 for TPX2 function during mitosis, TPX2 mutants defective in binding Aurora-A, ADD1, or Eg5 were generated (Fig 5A) and re-expressed into TPX2-depleted HeLa cells (Fig 5B). The TPX2 mutants retained their capability to associate with mitotic spindles (Fig EV5), suggesting that the deletion did not cause an overall disruption in their structures. TPX2 deletions at aa 1–43 and 711–747 have already been shown to abolish the interaction of TPX2 with Aurora-A and Eg5, respectively [17,44]. We show here that the deletion of TPX2 at aa 236–370 abolished its interaction with ADD1 but did not interfere with its interaction with Aurora-A (Fig 5C). TPX2 depletion caused two major defects in mitotic cells—reduced spindle length and multipolar spindles (Fig 5D–F). We further demonstrated that the multiple spindle poles caused by TPX2 depletion resulted from centriole splitting (Fig 6A–C). The TPX2 mutant defective in Aurora-A binding was able to restore the multipolar spindle defect but not spindle shrinkage (Fig 5D–F). In contrast, the TPX2 mutant defective in binding ADD1 or Eg5 was able to restore the spindle shrinkage defect but not multipolar spindles (Fig 5D–F) and centriole

splitting (Fig 6A–C). These results suggest that the interaction of TPX2 with Aurora-A is important for mitotic spindle assembly, while the interaction of TPX2 with both ADD1 and Eg5 is important for spindle pole integrity. Interestingly, we found that the inhibition of Eg5 by S-trityl-L-cysteine (STLC) significantly increased the formation of monopolar spindles and restored defect of multiple spindle poles induced by ADD1 depletion (Fig 7A and B). The bipolar spindle pole restored by STLC was intact, which contained a pair of centrioles (Fig 7C). In contrast, the spindle distortion and elongation induced by ADD1 depletion were not restored by STLC (Fig 7D and E). These data suggest that ADD1 depletion-induced multipolar spindle and centriole splitting may be Eg5-dependent.

# Discussion

Spindle pole integrity is essential for the fidelity of mitosis. During mitosis, spindle poles face traction and push forces driven by microtubule-based motors that move along spindle fibers. How do cells keep the integrity of spindle pole under such forces? One of the mechanisms may be via a process called centrosome maturation, which occurs at the onset of mitosis and is characterized by PCM expansion via the recruitment of PCM proteins, such as PCNT and γ-tubulin, to centrosomes [45,46]. PCM expansion may be necessary not only for organizing a large number of spindle microtubules but also for spindle pole integrity. In this study, we demonstrate that both ADD1 and TPX2 are PCM proteins (Figs 1, 2, and 4) and that their interaction is essential for spindle pole integrity (Figs 5 and 6). The phosphorylation of ADD1 at S726 is important for this interaction (Fig 4), which is most prominent during prophase and metaphase but decreased during telophase (Fig 1). The S726-phosphorylated ADD1 is organized into a rosette-like structure in the PCM, which is expanded during mitosis (Fig 2). Three possibilities exist for how the ADD1–TPX2 interaction contributes to spindle pole integrity. First, both ADD1 and TPX2 may serve as scaffold proteins within the PCM and their interaction may strengthen the scaffold for centriole engagement. Second, the interaction may regulate the behavior of microtubule-based motors and thereby regulate the balance between the traction and push forces that act on spindle poles. Third, the interaction may prevent the premature activation of separase during prometaphase [11], which cleaves the linker proteins between mother and daughter centrioles and thus leads to centriole disengagement [12,47]. However, the third possibility is less likely because ADD1 depletion did not induce premature separase activation (Appendix Fig S3).

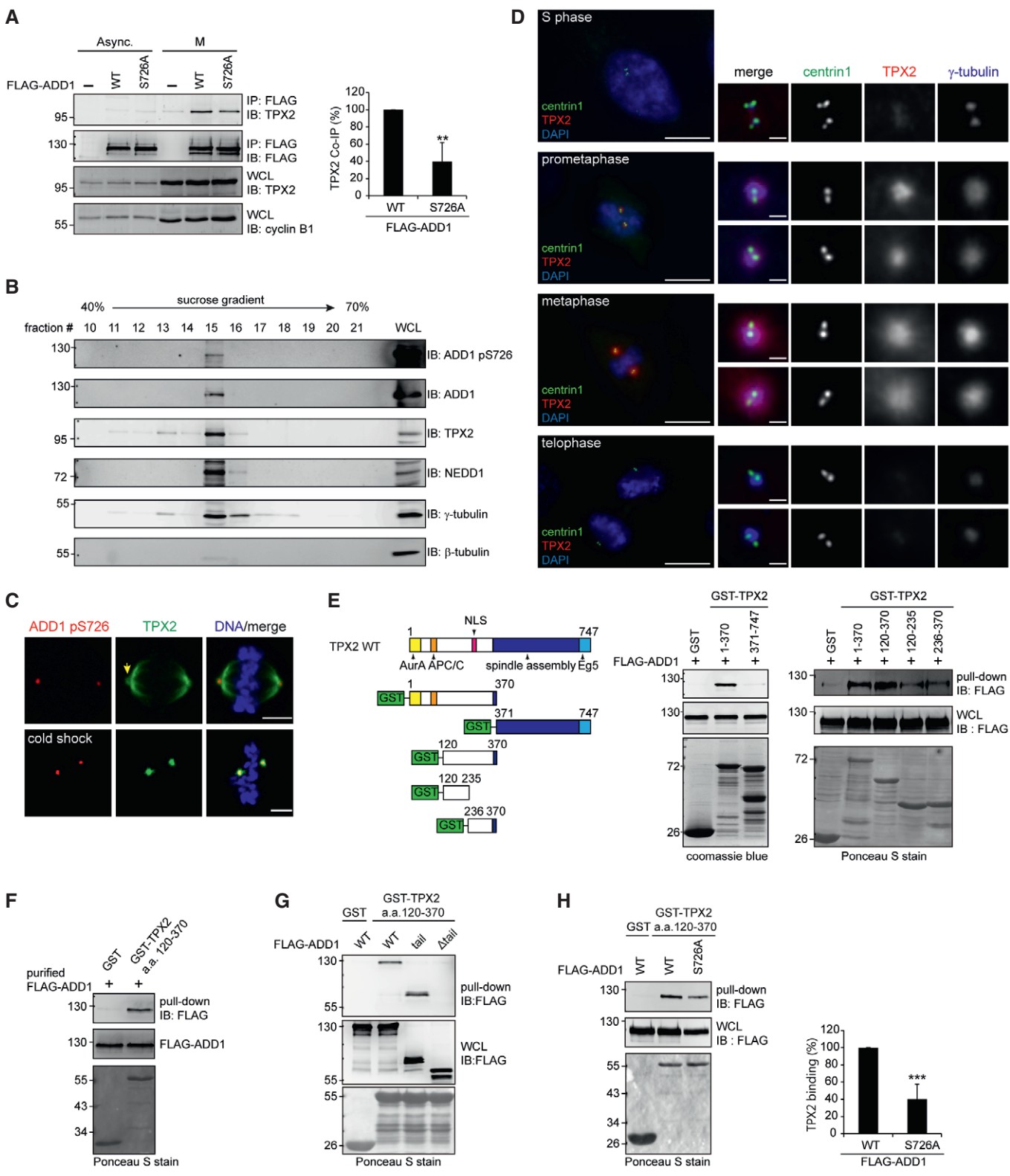

**Figure 4.**

The significance of TPX2 in mitotic spindle formation, which relies on its interactions with Aurora-A, Eg5, Myo10, and CLASP1, has been extensively studied [22–25,48]. TPX2 is important for both spindle assembly and spindle pole integrity [16–19], but the mechanism remains unclear. In this study, we show that TPX2 is recruited to mitotic centrosomes (Fig 4) and plays an essential role in

**Figure 4.  ADD1 phosphorylation at S726 is important for its interaction with TPX2.**

A   HeLa cells expressing FLAG-ADD1 WT or the S726A mutant remained asynchronized (Async.) or were synchronized at the M phase. Whole-cell lysates were incubated with anti-FLAG M2 affinity resins. The bound proteins were eluted from the resins with FLAG peptides and analyzed by immunoblotting (IB) with anti-FLAG and anti-TPX2 antibodies. WCL, whole-cell lysates.

B   Centrosomes were isolated from mitotic-arrested HeLa cells using discontinuous gradient ultracentrifugation. The fractions enriched with γ-tubulin were analyzed by immunoblotting with the indicated antibodies.

C   HeLa cells were either placed at 4°C for 30 min (cold shock) or left at 37°C before fixation and then stained for TPX2 (green), and ADD1 pS726 (red). The arrow indicates the spindle pole region. Scale bars, 5 μm.

D   RPE1 cells were placed at 4°C for 30 min before fixation and then stained for centrin1, TPX2, γ-tubulin, and DNA. Scale bars, 10 μm (main image) and 1 μm (zoomed images).

E   For the in vitro GST pull-down assay, immobilized GST-TPX2 fusion proteins were incubated with the cell lysates from HEK293 cells expressing FLAG-ADD1. The bound proteins were analyzed by immunoblotting (IB) with anti-FLAG antibody. The GST fusion proteins were visualized by Coomassie blue stain or Ponceau S stain.

F   FLAG-ADD1 was transiently expressed in HEK293 cells, affinity-purified by FLAG beads, and eluted with a FLAG peptide. Immobilized GST-TPX2 aa 120–370 fusion protein or GST alone (control) was incubated with purified FLAG-ADD1. The bound proteins were analyzed by immunoblotting (IB) with anti-FLAG antibody.

G   Immobilized GST-TPX2 aa 120–370 fusion protein or GST alone (control) was incubated with the cell lysates from HEK293 cells transiently expressing FLAG-ADD1, the tail domain, or the mutant with a deletion at the tail domain (Δtail). The bound proteins were analyzed by immunoblotting (IB) with anti-FLAG antibody.

H   Immobilized GST-TPX2 aa 120–370 fusion protein was incubated with the cell lysates from HEK293 cells transiently expressing FLAG-ADD1 WT or the S726A mutant. The bound proteins were analyzed by immunoblotting with anti-FLAG.

Data information: Values in (A and H) are means ± s.d. Data are from three independent experiments (A) or five independent experiments (H) and expressed as the percentage relative to the level of FLAG-ADD1 WT. **$P$ = 0.0091 and ***$P$ = 0.00006 (Student's $t$-test).
Source data are available online for this figure.

maintaining spindle pole integrity (Figs 5 and 6). TPX2 depletion induced shortened spindles and multipolar spindles resulting from centriole splitting (Figs 5 and 6). Our results indicated that while the interaction of TPX2 with Aurora-A is crucial for controlling spindle length (Fig 5), the interactions of TPX2 with ADD1 and Eg5 are essential for spindle pole integrity (Figs 5 and 6). TPX2 has been shown to bind Eg5 and suppress its motor activity during mitosis, which ensures proper bipolar spindle formation [25]. Interestingly, we found that the multipolar spindles and centriole splitting induced by ADD1 depletion were Eg5-dependent (Fig 7), rendering it possible that ADD1 binding may be necessary for TPX2 to interact with Eg5 and/or suppress Eg5 activity. Likewise, depletion of the dynein light chains also leads to centriole splitting, which can be rescued by inhibiting Eg5 [14]. This study and ours indicate that the mechanism which ensures the balance of motor-driven forces acting on spindle poles is important for spindle pole integrity. Loss or disturbance of this mechanism may lead to centriole splitting and thereby multipolar spindles.

Like its wild-type counterpart, ADD1 S726A mutant localizes to mitotic centrosomes, as analyzed by immunofluorescence staining and centrosome fractionation (Appendix Fig S4), indicating that the phosphorylation of ADD1 at S726 is not required for its localization to mitotic centrosomes. These data also suggest that ADD1 may be first recruited to mitotic centrosomes via a not-yet-known mechanism and then phosphorylated at S726 by a protein serine kinase in these subcellular compartments. However, the kinase responsible for phosphorylating ADD1 S726 during mitosis remains unknown. We have previously shown that the phosphorylation of ADD1 at S12 and S355 by CDK1 is required for its interaction with Myo10 and its association with mitotic spindles [42]. ADD1 interacts with Myo10 through its NH2-terminal head domain, which is essential for proper spindle assembly [42]. In this study, we show that the phosphorylation of ADD1 at S726 in the COOH-terminal tail domain facilitates its interaction with TPX2 and is crucial for spindle pole integrity. Mutating S726 does not affect ADD1 interaction with Myo10 (Appendix Fig S5). Likewise, ADD1 with mutations at both S12 and S355 retained its interaction with TPX2 (Appendix Fig S5). These data suggest that different subsets of ADD1 may preferentially interact with TPX2 and Myo10 through phosphorylation at different serine residues by different mitotic kinases. In this scenario, the ADD1$^{pS726}$ interacts with TPX2 at mitotic centrosomes for the spindle pole integrity, whereas ADD1$^{pS12/S355}$ interacts with Myo10 at spindle fibers for proper spindle assembly (Fig 7F).

The interaction of TPX2 with CLASP1, a microtubule rescue factor at kinetochores, is known to be important for the control of

**Figure 5.  ADD1–TPX2 interaction is important for bipolar spindle formation.**

A   HA-tagged TPX2 WT (HA-TPX2 WT) and the deletion mutants (HA-TPX2 Δ1–43, Δ236–370 and Δ711–747) were transiently expressed in HEK293 cells and then analyzed by immunoblotting (IB) with anti-TPX2 and anti-HA.

B   HeLa cells were infected with lentiviruses expressing shRNAs to TPX2 (sh-TPX2), or luciferase (sh-Luc.). HA-TPX2 WT or the mutants (Δ1–43, Δ236–370 and Δ711–747) were re-expressed in the cells whose endogenous TPX2 had been depleted. Equal amounts of whole-cell lysates were analyzed by immunoblotting (IB) with the indicated antibodies.

C   FLAG-tagged ADD1 (FLAG-ADD1) or Aurora-A (FLAG-AurA) was transiently co-expressed with HA-TPX2 or its mutants in HEK293 cells. FLAG-tagged proteins were immunoprecipitated (IP) by anti-FLAG, and the immunocomplexes were analyzed by immunoblotting with anti-FLAG or anti-HA antibody.

D   The cells were stained for TPX2 (green), α-tubulin (red), and DNA (blue). Scale bars, 5 μm.

E   The ratio of spindle length to cell diameter was measured (116–244 mitotic cells were counted in each group).

F   The percentage of multipolar spindles in the total counted mitotic cells was measured (299–1,323 mitotic cells were counted in each group).

Data information: Values (means ± s.d.) are from three independent experiments. *$P$ < 0.05, **$P$ < 0.01 (Student's $t$-test). For clarity, not all significances are indicated.
Source data are available online for this figure.

spindle length through regulating microtubule flux [48]. Phosphorylation of TPX2 at S121 and S125 by Aurora-A is important for its interaction with CLASP1 [48], but the region of TPX2 for CLASP1 binding remains unclear. In this study, we show that the aa 120–370

of TPX2 is sufficient for its interaction with ADD1. Due to the proximity of S121 and S125 to the ADD1-binding region, it is possible that the CLASP1-binding region may be proximal or overlapped with the ADD1-binding region. Thus, the possibility that ADD1 and

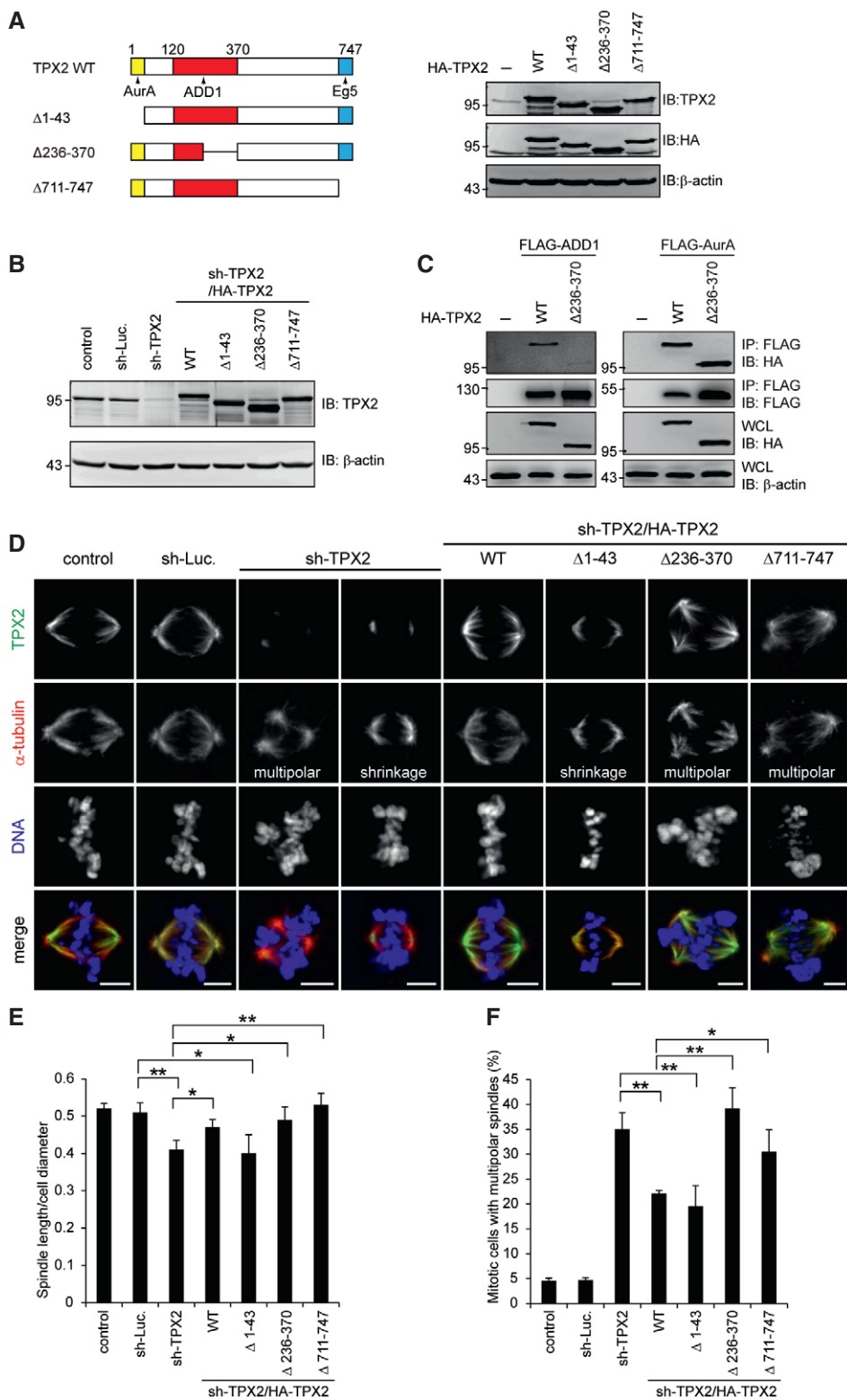

**Figure 5.**

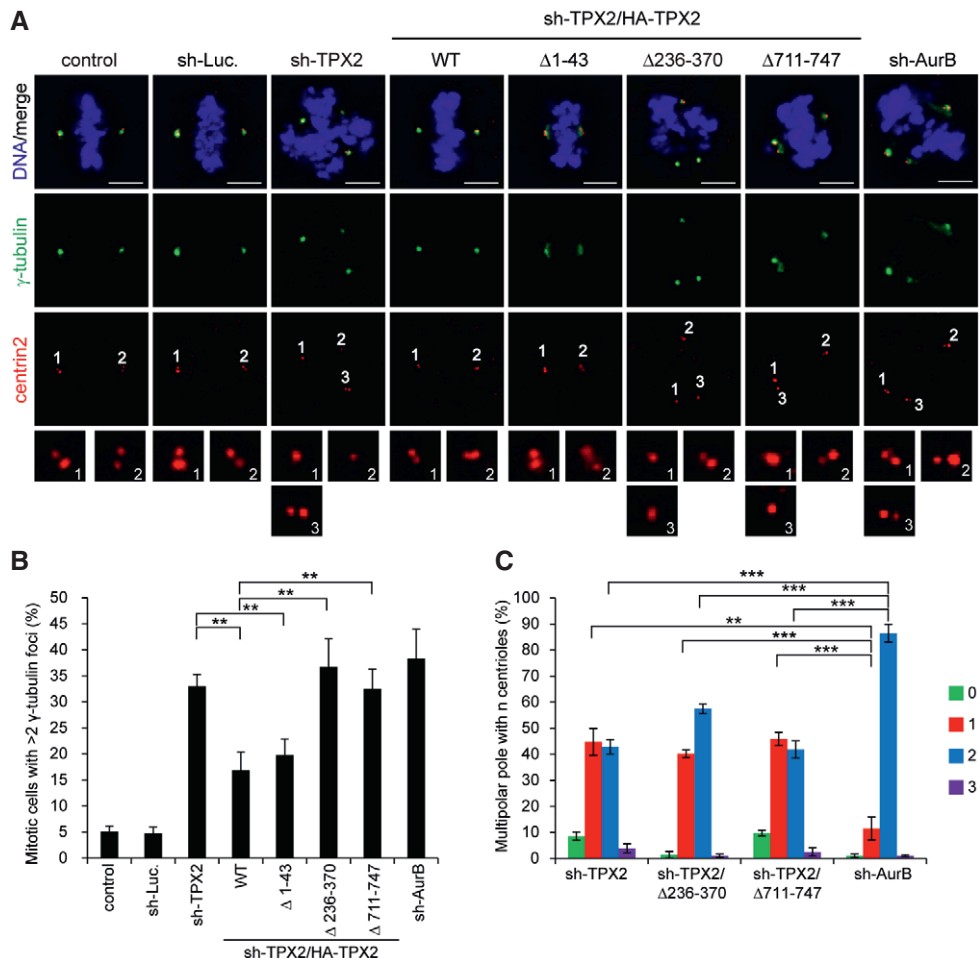

**Figure 6.  Interaction of TPX2 with ADD1 and Eg5 is important for spindle pole integrity.**

A  HeLa cells were infected with lentiviruses expressing shRNAs to TPX2 (sh-TPX2), or luciferase (sh-Luc.). HA-TPX2 WT or the mutants (Δ1–43, Δ236–370 and Δ711–747) were re-expressed in the cells whose endogenous TPX2 had been depleted. The cells were stained for γ-tubulin (green), centrin2 (red), and DNA (blue). The insets are magnification of the centrin2 signal at the indicated poles. Scale bars, 5 μm.

B  The percentage of multiple γ-tubulin foci in the total number of mitotic cells was measured (308–1,116 mitotic cells were counted in each group).

C  The percentage of poles with *n* centrioles found in cells with multiple γ-tubulin foci was assessed (746–1,239 poles were counted in each group).

Data information: Values (means ± s.d.) are from three independent experiments. **$P < 0.01$ and ***$P < 0.001$ (Student's *t*-test). For clarity, not all significances are indicated.

CLASP1 mutually affect their binding to TPX2 cannot be excluded. More experiments are needed to clarify this issue. However, the TPX2 mutant (with a deletion of aa 236–370) deficient in ADD1 binding retains S121 and S125, but fails to rescue the defect of multipolar spindle caused by TPX2 depletion (Fig 5), suggesting that CLASP1 binding, if there is any, is not sufficient to restore the defect.

Myo10 is an unconventional myosin that binds actin filaments (F-actin) and microtubules through a NH2-terminal motor domain and a myosin tail homology 4 domain, respectively [49,50]. This property makes Myo10 an unusual link between microtubules and actin cytoskeleton. Myo10 has been shown to be required for mitotic spindle assembly, spindle length control, and spindle positioning [22,50,51]. The function of Myo10 in mitotic spindle formation relies on its interaction with TPX2 [22] and ADD1 [42]. ADD1 is known to bind the motor domain of Myo10 [42], whereas how Myo10 and TPX2 interact is unclear. Moreover, because the

TPX2–ADD1 interaction is important for spindle pole integrity, the role of Myo10 in this regard requires further studies.

ADD1 is an actin-binding protein; however, it is unclear whether F-actin is involved in the functions of ADD1 during mitosis. The phosphorylation of ADD1 at S726 within the MARCKS-related motif has been shown to decrease the binding of ADD1 to spectrin and F-actin [10,34]. In addition, F-actin has not been detected in centrosomes. Therefore, the function of S726-phosphorylated ADD1 in mitotic centrosomes is likely F-actin-independent. In contrast, the function of ADD1 in spindle assembly is more likely F-actin-dependent. Myo10 is the ADD1 binding partner on mitotic spindles [42] and its role in spindle positioning and spindle length control is F-actin-dependent [22]. In fact, the spindle F-actin has been shown to surround mitotic spindles during early *Xenopus* embryo mitosis [22]. Arp2/3 protein complex, which is involved in the nucleation and assembly of branched actin filaments, has been shown to

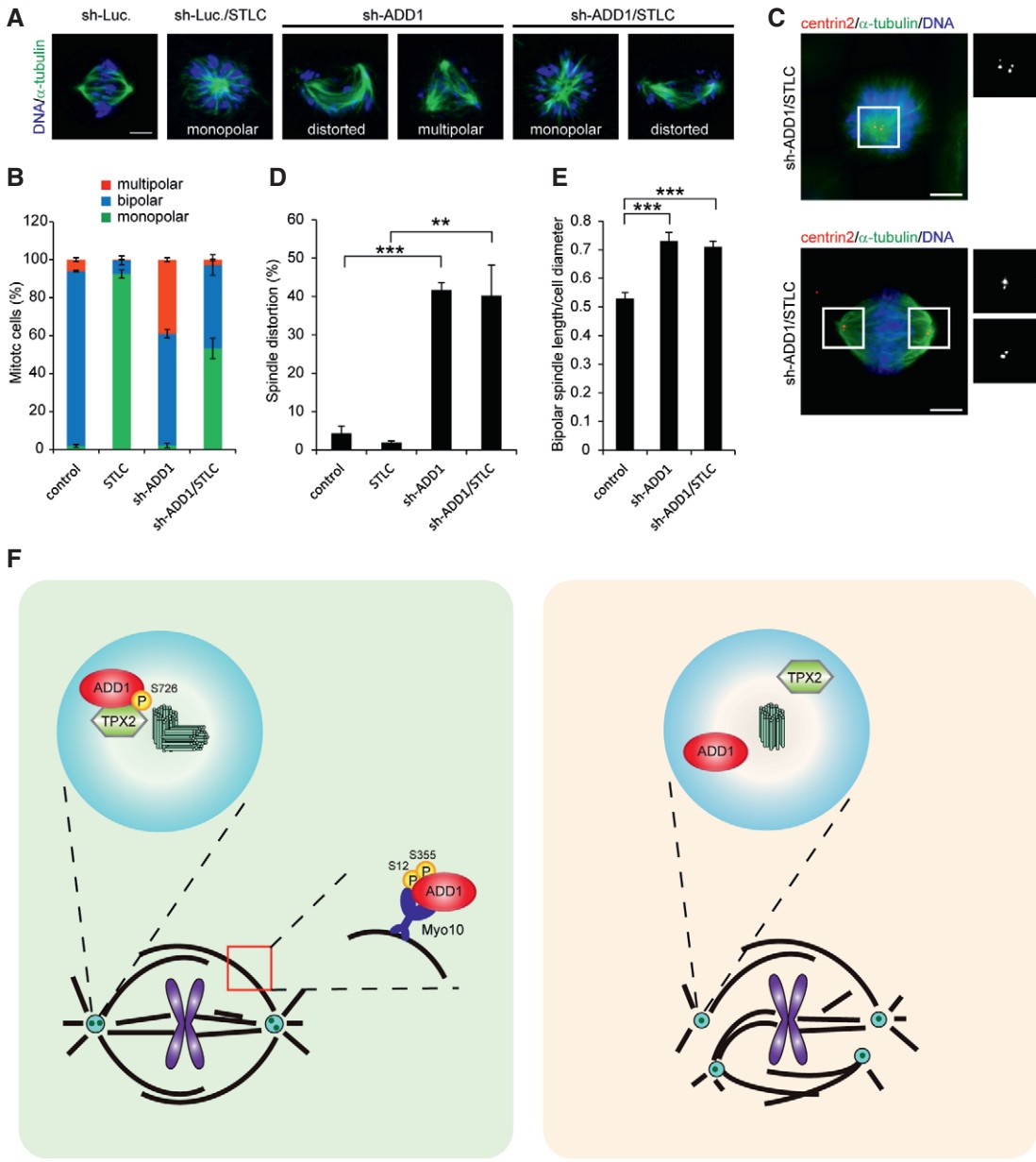

**Figure 7. Spindle multipolarity induced by ADD1 depletion is Eg5-dependent.**

A   HeLa cells were infected with lentivirus expressing shRNA to ADD1 (sh-ADD1). The cells were treated with 2 μM STLC for 2 h or left untreated and then stained for α-tubulin and DNA. Scale bars, 5 μm.

B   The percentage of monopolar, bipolar, and multipolar spindles in the total number of mitotic cells was measured (491–1,508 mitotic cells were counted in each group).

C   The cells with ADD1 depletion were treated with 2 μM STLC for 2 h and then stained for α-tubulin, centrin2, and DNA. Scale bars, 5 μm.

D   The percentage of distorted spindles in the total number of mitotic cells counted was measured (491–1,508 mitotic cells were counted in each group).

E   The ratio of spindle length to cell diameter was measured (106–110 mitotic cells were assessed in each group).

F   Model for the role of ADD1 in mitotic spindle formation. Left, we show in this study that the phosphorylation of ADD1 at S726 in the COOH-terminal tail domain facilitates its interaction with TPX2 and is crucial for spindle pole integrity. The phosphorylation of ADD1 at S12 and S355 by CDK1 is required for its interaction with Myo10 and its association with mitotic spindles, which is essential for proper spindle assembly (Chan *et al* [42]). Right, failure of the ADD1–TPX2 interaction causes centriole splitting and thereby multipolar spindles, which may be Eg5-dependent.

Data information: Values in (B, D, and E) are means ± s.d. Data are from three independent experiments. Statistical significance of differences is assessed with a Student's *t*-test: for (D), **$P = 0.0011$ and ***$P = 0.00002$; for (E), ***$P < 0.001$.

participate in the formation of the spindle F-actin [52,53]. In summary, we demonstrate that ADD1 phosphorylation at S726 is important for its interaction with TPX2 and for spindle pole integrity. This work not only unveils a novel role for ADD1 in maintaining spindle pole integrity but also highlights the significance of ADD1–TPX2 interactions during mitosis.

# Materials and Methods

## Materials

The rabbit polyclonal anti-ADD1 pS726 (sc-16736), anti-ADD1 (sc-25731), anti-Aurora-B (sc-25426), and anti-centrin2 (sc-27793) antibodies and mouse monoclonal anti-cyclin B1 (sc-245), anti-PLK1 (sc-17783), anti-GFP (sc-9996), anti-His (sc-8036), anti-c-Nap1 (sc-390540), anti-SAS6 (sc-81431), and anti-β-tubulin (sc-5274) antibodies were purchased from Santa Cruz Biotechnology. The rabbit polyclonal anti-FLAG (F7425) and anti-γ-tubulin (T3559) antibodies and mouse anti-β-actin (A5441), anti-γ-tubulin (T6557), anti-α-tubulin (T9026), and anti-FLAG (F3165) antibodies were purchased from Sigma-Aldrich. The mouse monoclonal anti-PLK1 pT210 (ab39068), anti-Aurora-A (ab13824), anti-NEDD1 (ab57336), mouse anti-pericentrin (ab28144), anti-TPX2 (ab32795), and anti-separase (ab16170) antibodies were purchased from Abcam. The rabbit monoclonal anti-Aurora-A pT288 (#3079) was purchased from Cell Signaling Technology. The rabbit polyclonal anti-TPX2 (NB500-179) antibody was purchased from Novus Biologicals. The mouse monoclonal anti-centrin1 (#04-1624) antibody was purchased from EMD Millipore. The rabbit polyclonal anti-α-tubulin (GTX113098) antibody was purchased from GeneTex. The mouse monoclonal anti-HA (MMS-101P) antibody was purchased from Covance. Nocodazole was purchased from Sigma-Aldrich. S-trityl-L-cysteine was purchased from Santa Cruz Biotechnology. RO3306 was purchased from Enzo Life Sciences. G418, puromycin, and hygromycin were purchased from Merck.

## Plasmids and mutagenesis

The FLAG-Eg5 plasmid was kindly provided by Wen H. Shen (Weill Cornell Medical College, New York, NY). The pCMV2-FLAG-Aurora-A plasmid was kindly provided by Pei-Jung Lu (National Cheng-Kung University, Tainan, Taiwan). The pEGFP-PLK1 plasmid was kindly provided by Chin Li (National Chung-Cheng University, Chia-Yi, Taiwan). The plasmids pCMV-3Tag-3A-ADD1 WT and pCMV-3Tag-3A-ADD1 S726A for FLAG-ADD1 WT and S726A were constructed in our laboratory and described previously [42]. The S726D point mutation in pCMV-3Tag-3A-ADD1 was generated by using a site-directed mutagenesis kit (QuikChange, Agilent Technologies) and was confirmed by dideoxy DNA sequencing. To generate the S726D mutation in pCMV-3Tag-3A-ADD1, the forward primer was 5′-GAAGAAGTTCCGTACCCCGGACTTTCTGAAGAA GAACAAG-3′ and the reverse primer was 5′-CTTGTTCTTCTTC AGAAAGTCCGGGGTACGGAACTTCTTC-3′. To express the FLAG-ADD1 WT, or S726A mutant by lentiviral infection, the corresponding cDNAs were PCR-amplified using pCMV-3Tag-3A-ADD1 WT, or S726A as the template with the primers listed below and then subcloned into the lentiviral vector pLAS3w.Pneo (National RNAi Core Facility, Academia Sinica, Taipei, Taiwan). The forward primer was 5′-CTCGCTAGCATGAATGGTGATTCTCGTGCT-3′, and the reverse primer was 5′-CCGTTTAAACCTATTTATCGTCATCATCTTT GTAG-3′. To express the FLAG-ADD1 S726D mutant by lentiviral infection, the corresponding cDNA was PCR-amplified using pCMV-3Tag-3A-ADD1 S726D as the template with the primers listed below and then subcloned into the lentiviral vector pLAS3w.Phyg (National RNAi Core Facility, Academia Sinica, Taipei, Taiwan).

The forward primer was 5′-CTCGCTAGCATGAATGGTGATTCT CGTGCT-3′, and the reverse primer was 5′-CCGTTTAAACCTATT TATCGTCATCATCTTTGTAG-3′.

Plasmid harboring the cDNA for full-length TPX2 (Homo sapiens) was purchased from Transomic Technologies. The full-length human TPX2 cDNA (aa 1–747) was amplified by PCR using the primers listed below and then cloned into the pcDNA3.1(+)-HA3 vector. The forward primer was 5′-AATAGATCTATGTCACAAGT TAAAAGCTCTTATTCC-3′, and the reverse primer was 5′-AATGTC GACTTAGCAGTGGAATCGAGTGGAGAATTTG-3′. To construct the pcDNA3.1(+)-HA3 plasmids encoding HA-TPX2 aa 1–43, aa 236–370, and aa 711–747, the corresponding cDNA fragments were PCR-amplified using pcDNA3.1(+)-HA3-TPX2 as the template with the primers listed below and then subcloned into the pcDNA3.1(+)-HA3 vector. For HA-TPX2 1–43, the forward primer was 5′-AATGGATC CAAGTTACTGGGGAAGAATGGAACTGGAGGG-3′ and the reverse primer was 5′-AATGCGGCCGCTTAGCAGTGGAATCGAGTGGAGAA TTTG-3′. For HA-TPX2 236–370, the forward primers were 5′-AATG GATCCATGTCACAAGTTAAAAGCTCTTATTCC-3′ and 5′-AATAAGC TTGTACTGCAAACCAAACACCGTGC-3′ and the reverse primers were 5′-AATAAGCTTGAATTCTTCATTCTTTTTCCG-3 and 5′-AATG GATCCTTAGCAGTGGAATCGAGTGGAGAATTTG-3′. For HA-TPX2 aa 711–747, the forward primer was 5′-AATGGATCCATGTCA CAAGTTAAAAGCTCTTATTCC-3′ and the reverse primer was 5′-AATGCGGCCGCTTATTCTCTCCGTAGCCTGGCCAGCTCCTC-3′. To express HA-TPX2 WT, aa 1–43, aa 236–370, and aa 711–747 by lentiviral infection, HA-TPX2 WT, aa 1–43, aa 236–370, and aa 711–747 were cloned into the NheI and PmeI sites of the pLAS3w.Pneo plasmid. To generate HA-TPX2 resistant to TPX2-specific shRNA, three nucleotides, T195C, T198C, and G201A, were substituted by site-directed mutagenesis in the pLAS3w.Pneo-HA-TPX2 construct with the forward primer 5′-CTAACCTCCAACAAGCTATTG-3′ and the reverse primer 5′-CAATAGCTTGTTGGAGGTTAG-3′. To construct the plasmids encoding GST-tagged TPX2 aa 1–370, aa 120–370, aa 120–235, aa 236–370, and aa 371–747, cDNAs were PCR-amplified using pcDNA3.1(+)-HA3-TPX2 as the template with the primers listed below and then subcloned into the pGEX-2T vector. For GST-TPX2 aa 1–370, the forward primer was 5′-AATG GATCCATGTCACAAGTTAAAAGCTCTTATTCC-3′ and the reverse primer was 5′-AATGGATCCTTAAGGAGTCTGTGGGTCTCTGCAA ATC-3′. For GST-TPX2 aa 120–370, the forward primer was 5′-AATGGATCCAGATCTCTTAGGCTTTCTGCT-3′ and the reverse primer was 5′-AATGGATCCTTAAGGAGTCTGTGGGTCTCTGCAAA TC-3′. For GST-TPX2 aa 120–235, the forward primer was 5′-AATG GATCCAGATCTCTTAGGCTTTCTGCT-3′ and the reverse primer was 5′-GAATTCTTCATTCTTTTTCCGCAT-3′. For GST-TPX2 aa 236–370, the forward primer was 5′-AATGGATCCAAGAAACTTGC TCTGGCTGGA-3′ and the reverse primer was 5′-AATGGATCCTT AAGGAGTCTGTGGGTCTCTGCAAATC-3′. For GST-TPX2 aa 371–747, the forward primer was 5′-AATGGATCCGTACTGCAAACCAAA CACCGTGCACGG-3′ and the reverse primer was 5′-AATGGATCCT TAGCAGTGGAATCGAGTGGAGAATTTG-3′.

## shRNA and lentiviral production

The lentiviral expression system, consisting of the pLKO-AS1-puromycin (puro) plasmid encoding shRNAs, the pLAS3w.Phyg plasmid, and the pLAS3w.Pneo plasmid, was obtained from the National

RNAi Core Facility (Academia Sinica, Taiwan). The target sequences for ADD1 were 5′-GCAGAATTTACAGGACATTAA-3′ (#1) and 5′-GATAGGACTTTCCACCTGATT-3′ (#2). The target sequence for TPX2 was 5′-CTAATCTTCAGCAAGCTATTG-3′. The target sequence for Aurora-B was 5′-CAGCCGAGTCCTCCGGAAAGAG-3′. For FLAG-ADD1 WT, S726A, and HA-TPX2 expression, FLAG-ADD1 and HA-TPX2 cDNA were amplified by PCR and then subcloned into the pLAS3w.Pneo vector. For FLAG-ADD1 S726D expression, FLAG-ADD1 S726D cDNA was amplified by PCR and then subcloned into the pLAS3w.Phyg vector. Lentiviral production was performed as described previously [42].

### Cell culture and cell cycle synchronization

HeLa and HEK293 cells were obtained from American Type Culture Collection and maintained in DMEM supplemented with 10% fetal bovine serum (FBS; Invitrogen). RPE1 cells were cultured in DMEM/F-12 (1:1) with 10% FBS. All cells were cultured at 37°C in a humidified atmosphere of 5% $CO_2$ and 95% air. For transient transfections, HEK293 cells ($10^6$) were seeded in 6-cm culture dishes. After 24 h, the cells were incubated with the mixture of plasmid DNA (2 µg) and Lipofectamine (6 µl; Invitrogen) for 6 h and allowed to grow for an additional 24 h. To synchronize HeLa cells in the mitotic phase, the cells were treated with 200 ng/ml nocodazole for 16–18 h. To collect mitotic cells, the rounded mitotic cells were shaken from the plate and collected by centrifugation. To release the mitotic cells to the G1 phase, they were collected and grown for an additional 4 h.

### Centrosome purification

Centrosomes were isolated from mitotic-arrested HeLa cells using discontinuous gradient ultracentrifugation as described previously [54]. In brief, the cell pellets were washed with 1× PBS and 0.1× PBS/8% sucrose. Cells were resuspended with lysis buffer (1 mM Tris pH 8.0, 0.1% (v/v) 2-mercaptoethanol, 0.5% (v/v) NP-40, and 0.5 mM $MgCl_2$) containing protease inhibitor cocktail (Roche). The suspension was shaken slowly for 30 min at 4°C and spun at 2,500 × g for 10 min. The supernatant was added to 10 mM PIPES and 1 µg/ml DNase. After incubation for 30 min on ice, the mixture was gently underlaid with a 60% sucrose solution and centrifuged at 10,000 × g for 30 min. The obtained centrosomal suspension was vortexed, loaded onto a discontinuous sucrose gradient (70, 50, and 40% sucrose solutions from the bottom), and centrifuged at 120,000 × g for 1 h. Fractions were collected from the top, diluted with 1× PBS, and centrifuged at 20,000 × g for 10 min. The supernatants were removed, and the centrosomes were resuspended in SDS sample buffer.

### Immunoblotting and immunoprecipitation

To prepare whole-cell lysates, cells were lysed with 1% NP-40 lysis buffer (1% NP-40, 20 mM Tris–HCl pH 8.0, 137 mM NaCl, 10% glycerol, and 1 mM $Na_3VO_4$) containing protease inhibitor cocktail (Roche). To precipitate FLAG-ADD1, the cells were lysed in TBS buffer (1% Triton X-100, 50 mM Tris–HCl pH 7.4, and 150 mM NaCl) containing protease inhibitors and subjected to centrifugation at 19,200 × g at 4°C for 10 min. The cell lysates were incubated

with anti-FLAG M2 affinity resins (Sigma-Aldrich) for 16 h according to the manufacturer's instructions. FLAG-ADD1 proteins were eluted from the resins with 100 µM 3× FLAG peptide (Sigma-Aldrich), boiled for 3 min in SDS sample buffer, subjected to SDS–polyacrylamide gel electrophoresis, and transferred to nitrocellulose (Schleicher and Schuell). Immunoblotting was performed with the indicated antibodies using the Western Chemiluminescent HRP Substrate (Immobilon; EMD Millipore) for detection. Chemiluminescent signals were detected by a luminescence imaging system (Fuji LAS-4000 mini).

### *In vitro* pull-down assay

GST-TPX2 fusion proteins were expressed in *Escherichia coli* (BL21) and purified on glutathione agarose beads (GE Healthcare). Immobilized GST-TPX2 fusion proteins were incubated with 25 µg of lysates from HEK293 cells expressing FLAG-tagged ADD1 (293/FLAG-ADD1) at 4°C for 60 min. The beads were washed three times with TBS buffer and eluted with SDS sample buffer. The bound proteins were analyzed by immunoblotting with anti-FLAG antibody. The GST fusion proteins were visualized by Ponceau S stain or Coomassie blue stain.

### Immunofluorescence staining, super-resolution imaging, and image analysis

Cells were fixed with methanol at −20°C for 10 min or 3% paraformaldehyde in 90% methanol at −20°C for 30 min and then permeabilized with 0.1% Triton X-100 for 10 min at room temperature. The fixed cells were stained with primary antibodies at room temperature for 2 h and then incubated with Alexa Fluor 488-, 546-, or 680-conjugated secondary antibodies (Invitrogen) for 2 h. Coverslips were mounted on the slides with mounting medium (Anti-Fade Dapi-Fluoromount-G, Southern Biotech). The images in Figs 1B, C, 3B, D, and 4C were acquired using a laser-scanning confocal microscope imaging system (Carl Zeiss LSM 510) with a Plan Apochromat 100×/NA 1.4 oil immersion objective (Carl Zeiss). The images in Figs 1D, E, 4D, 5D, 6A, and 7A, C were acquired on an upright microscope (Axio imager. M2 ApoTome2 system, Carl Zeiss) equipped with a Plan Apochromat 100×/NA 1.4 oil immersion objective and a camera (ORCA-Flash4.0 V2; Hamamatsu).

Structured illumination microscopy super-resolution images were acquired using a Zeiss ELYRA PS.1 system equipped with a Plan Apochromat 63×/NA 1.4 oil immersion objective and ZEN software (Institute of Molecular Biology, Academia Sinica). dSTORM super-resolution imaging was performed using a Nikon Ti-E microscope. The 637 nm (OBIS 637 LX, Coherent) and 405 nm (OBIS 405 LX, Coherent) laser sources were introduced into the samples through the back focal plane of a 100×/1.49 NA oil immersion objective (CFI Apo TIRF, Nikon) for the wide-field illumination. The 637 nm light was operated at an intense power density of 3–5 kW/cm$^2$ for both the exciting and quenching fluorophores (Alexa 647), while the weak 405 nm light was gradually added for fluorophore conversion from a dark state to an excitable state. Fluorescence emission was collected by the same objective and then projected onto an EMCCD (Evolve 512 Delta, Photometrics), after passing through a bandpass filter (700/75, Chroma). During the acquisition, individual single-molecule emission peaks as well as fiducial markers (TetraSpeck,

Thermo Fisher) for drift correction were recorded in real-time and localized using the MetaMorph Super-resolution Module (Molecular Devices); samples were incubated in an imaging buffer containing an oxygen scavenging system at pH 8.0 (60–100 mM mercaptoethylamine (MEA), 0.5 mg/ml glucose oxidase, 40 μg/ml catalase, and 10% (w/v) glucose; Sigma-Aldrich). In total, 10,000–20,000 frames were typically acquired at a rate of 50 fps to reconstruct each dSTORM image. The super-resolution images shown in the figures were cleaned with a Gaussian filter with a radius of 0.7–1 pixel. The images were cropped with Photoshop CS6 (Adobe) and assembled with Illustrator CS6 (Adobe).

## Live-cell imaging

HeLa S3 cells stably expressing GFP-tubulin [55] were infected with lentiviruses encoding shRNAs to ADD1, TPX2, Myo10, or luciferase. Cells were maintained in a microcultivation system with temperature and $CO_2$ control devices (Carl Zeiss). The cells were monitored on an inverted microscope (Axio Observer; Carl Zeiss) using a LD Plan-NEOFLUAR 20×, NA 0.4 objective. Images were captured every 3 min for 18 h using a digital camera (ORCA-Flash4.0 V2; Hamamatsu) and were analyzed by ZEISS ZEN2 image software.

## Mass spectrometry

FLAG-ADD1 was transiently overexpressed in HEK293 cells, and the cells were synchronized in the G2/M phase by nocodazole. FLAG-ADD1 was bound to anti-FLAG M2 affinity gel (Sigma-Aldrich) and was eluted by 3× FLAG peptides (100 μM; Sigma-Aldrich). Mass spectrometry to identify the phosphorylation sites was performed as described previously [42].

## Statistics

Significance was determined by unpaired Student's $t$-tests for two samples and two-way ANOVAs for grouped data. Error bars represent standard deviation (s.d.). The significance levels are indicated by asterisks: *$P < 0.05$, **$P < 0.01$, and ***$P < 0.001$.

**Expanded View** for this article is available online.

## Acknowledgements

Wen-Hsin Hsu carried out her thesis research under the auspices of the Ph.D. Program in Tissue Engineering and Regenerative Medicine, National Chung Hsing University and National Health Research Institutes, Taiwan. We are grateful to Dr. Jyh-Lyh Juang at the National Health Research Institutes for helpful discussion. This work was supported by the Ministry of Science and Technology, Taiwan (Grant Number 105-2320-B-005-005-MY3, 106-2320-B-005-011-MY3, and 107-2923-B-005-002-MY3), and the Cancer Progression Research Center, National Yang-Ming University, from the Featured Areas Research Center Program within the framework of the Higher Education Sprout Project by the Ministry of Education (MOE) in Taiwan.

## Author contributions

H-CC designed the research project. W-HH and W-YL performed the experiments and analyzed the data; Y-MH and C-CL contributed to mass spectrometry; W-JW contributed to epifluorescence microscopy; J-CL contributed to super-resolution microscopy; W-HH and H-CC wrote the manuscript.

## Conflict of interest

The authors declare that they have no conflict of interest.

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
