## [Review Process File · EMBO Reports]

Adducin-1 is essential for spindle pole integrity through its interaction with TPX2

Wen-Hsin Hsu, Won-Jing Wang, Wan-Yi Lin, Yu-Min Juang, Chien-Chen Lai, Hong-Chen Chen

Review timeline:

Submission date:	7 December 2017
Editorial Decision:	18 January 2018
Revision received:	17 April 2018
Editorial Decision:	18 May 2018
Revision received:	21 May 2018
Accepted:	28 May 2018

Transaction Report:

1st Editorial Decision

18 January 2018

Thank you for the submission of your research manuscript to our journal. We have now received the full set of referee reports that is copied below.

As you will see, the referees acknowledge that the findings are potentially interesting. However, they all point out several technical concerns and have a number of suggestion for how the study should be strengthened. The major concerns regard potential secondary effects on centriole splitting caused by a mitotic delay and secondary effects of Plk1 inhibition on centrosome maturation, a potential role for Myo10 in the ADD1/TPX2 complex, missing control experiments and quantification and also the acknowledgement of earlier literature that found TPX2 at the centrioles.

From these comments it is clear that a significant revision is required before the manuscript becomes potentially suitable for publication in EMBO reports. However, given the potential interest of your findings and the overall supportive comments, I would like to give you the opportunity to address the concerns and would be willing to consider a revised manuscript with the understanding that the referee concerns must be fully addressed and their suggestions (as detailed above and in their reports) taken on board.

Should you decide to embark on such a revision, acceptance of the manuscript will depend on a positive outcome of a second round of review and I should also remind you that it is EMBO reports policy to allow a single round of revision only and that, therefore, acceptance or rejection of the manuscript will depend on the completeness of your responses included in the next, final version of the manuscript.

Revised manuscripts should be submitted within three months of a request for revision; they will otherwise be treated as new submissions. Please contact us if a 3-months time frame is not sufficient for the revisions so that we can discuss the revisions further.

Supplementary/additional data: The Expanded View format, which will be displayed in the main

HTML of the paper in a collapsible format, has replaced the Supplementary information. You can submit up to 5 images as Expanded View. Please follow the nomenclature Figure EV1, Figure EV2 etc. The figure legend for these should be included in the main manuscript document file in a section called Expanded View Figure Legends after the main Figure Legends section. Additional Supplementary material should be supplied as a single pdf labeled Appendix. The Appendix includes a table of content on the first page, all figures and their legends. Please follow the nomenclature Appendix Figure Sx throughout the text and also label the figures according to this nomenclature. For more details please refer to our guide to authors.

Regarding data quantification, please ensure to specify the number "n" for how many experiments were performed, the bars and error bars (e.g. SEM, SD) and the test used to calculate p-values in the respective figure legends. Please also include scale bars in all microscopy images.

We now strongly encourage the publication of original source data with the aim of making primary data more accessible and transparent to the reader. The source data will be published in a separate source data file online along with the accepted manuscript and will be linked to the relevant figure. If you would like to use this opportunity, please submit the source data (for example scans of entire gels or blots, data points of graphs in an excel sheet, additional images, etc.) of your key experiments together with the revised manuscript. Please include size markers for scans of entire gels, label the scans with figure and panel number, and send one PDF file per figure or per figure panel.

- a complete author checklist, which you can download from our author guidelines (<http://embor.embopress.org/authorguide#revision>). Please insert page numbers in the checklist to indicate where the requested information can be found.
 - a letter detailing your responses to the referee comments in Word format (.doc)
 - a Microsoft Word file (.doc) of the revised manuscript text
 - editable TIFF or EPS-formatted figure files in high resolution
- (In order to avoid delays later in the publication process, please check our figure guidelines before preparing the figures for your manuscript:
http://www.embopress.org/sites/default/files/EMBOPress_Figure_Guidelines_061115.pdf)
- a separate PDF file of any Supplementary information (in its final format)
 - all corresponding authors are required to provide an ORCID ID for their name. Please find instructions on how to link your ORCID ID to your account in our manuscript tracking system in our Author guidelines (<http://embor.embopress.org/authorguide>).

As part of the EMBO publication's Transparent Editorial Process, EMBO reports publishes online a Review Process File to accompany accepted manuscripts. This File will be published in conjunction with your paper and will include the referee reports, your point-by-point response and all pertinent correspondence relating to the manuscript.

I look forward to seeing a revised version of your manuscript when it is ready. Please let me know if you have questions or comments regarding the revision.

REFeree REPORTS

Referee #1:

Review of Adducin-1

This paper describes a phosphorylation-dependent role for the actin-binding protein ADD1 at the mitotic spindle poles, which the authors at least in part attribute to an interaction between ADD1 and TPX2.

Key observations:

1. Phospho-Ser726 ADD1 is present at the mitotic centrosome
2. Phosphorylation at Ser726 is required for preserving spindle bipolarity by ADD1
3. ADD1 and TPX2 interact
4. The ADD1-binding domain of TPX2 is required for normal spindle length

The manuscript contains high quality data and provides convincing evidence for the centrosomal localization of Phospho-Ser726 ADD1. However, there are a number of issues that prevent me from recommending the paper for publication in its current state. Some of these are technical issues, whereas others question the conceptual advance made by the authors.

Major points:

1. The centriole splitting phenotype shown in Fig. 3 could arise from the extended mitotic delay previously reported in ADD1-depleted cells by Chan et al., 2014 JCB. The authors should perform time-lapse microscopy on cells expressing fluorescent reporters that allow visualization of spindle morphology. This would reveal if multipolarity occurs upon entry into mitosis or only later during the mitotic arrest. They could compare/contrast these to Myo10- and TPX2-depleted cells. Also, the authors suggest that the role of TPX2 in spindle pole integrity has not been elucidated, but this is not the case and they should cite papers that described centriole splitting in TPX2-depleted cells previously (i.e. de Luca et al., Cell cycle, 2006). The phenomenon was already described in Garrett et al., 2002, which the authors cite but only for describing its role in spindle bipolarity.
2. Although the authors show that ADD1 and TPX2 co-sediment in sucrose fractionation, it would be important to demonstrate that endogenous ADD1 co-immunoprecipitates with endogenous TPX2. Currently, one partner is always overexpressed and there is no *in vitro* data to support direct interaction between the two proteins. If antibodies are not suitable for immunoprecipitation, a tag could be knocked into the ADD1 locus.
3. From the manuscript it is unclear why the authors tested whether ADD1 could bind TPX2. There should be a clearer explanation in the text, but for now I can only assume it is because TPX2 was previously shown to bind Myo10, and Myo10 is also a binding partner of ADD1. If this is so, it is surprising that there is no attempt in the paper to investigate whether these proteins form a ternary complex. Does Ser726 phosphorylation interfere with binding of ADD1 to Myo10? How do Ser12 and Ser355 phosphorylations affect TPX2 binding? What is the hierarchy of centrosome and spindle recruitment by Myo10/TPX2/ADD1? Does ADD1 also co-immunoprecipitate with Aurora-A? Does ADD1 overexpression/depletion affect Aurora-A binding to the mitotic spindle (i.e. by sequestration of TPX2)? Addressing these questions would provide mechanistic insight and improve the novelty and impact of the study.
4. Related to point 3, the authors need to place their finding that TPX2 and ADD1 are in a protein complex into a broader picture, as it is hardly surprising that two proteins with a common binding partner might interact. The model in Fig. 8 should be modified accordingly.
5. The authors suggest that Ser726 of ADD1 is phosphorylated by PLK1. Based on Fig. 4A the BI2536 treatment only slightly reduced levels of phospho-T210 PLK1, which may explain why there is only a small reduction in phospho-Ser726. One would expect a more complete response to this highly specific inhibitor. Centrosomal levels of ADD1 are clearly affected by BI2536 treatment, but PLK1 inhibition is known to disrupt the PCM, so this may not be a direct effect. Phosphorylations of ADD1 could also be mapped by MS in absence/presence of BI2536. These are

not essential experiments for the paper as a whole, but if not performed, it might be wiser to remove this figure.

6. The term 'spindle pole integrity' is used in different contexts in the text. For instance, in the last paragraph the authors use it to describe shorter spindles. These should be clarified.

7. Statistical tests are needed for all experimental groups in Fig. 6 E and F. Based on Fig. 6E binding by TPX2 to Eg5 or ADD1 is redundant in terms of spindle length maintenance as either deletion construct is capable of rescuing spindle length in shTPX2-treated cells, so the authors should not use the term "essential". Since interaction of TPX2 with CLASP is also implicated in spindle length regulation, I wonder if the authors excluded the possibility that CLASP and ADD1 interact the same domain of TPX2.

Minor points:

1. Labelling in figures should be more complete (e.g. Fig5E: include which protein/fragment is tagged with what).
2. Fig. 4A: second blot, correct IB: Auroa A to Aurora A.

Referee #2:

In this manuscript, the authors show that phosphorylation at Serine 726 of ADD1, a membrane- and actin-binding protein, is important to maintain centrosome and spindle pole integrity during mitosis in HeLa cells. They provide evidence that S726 is phosphorylated by Plk1 and dephosphorylated by PP1 and that this phosphorylation enhances the interaction between ADD1 and TPX2. The results suggest a model in which pS726 ADD1, TPX2, and Eg5 cooperatively prevent centriole splitting during mitosis. Overall the results are interesting and novel. However, the presentation needs work, with additional controls and quantification, as well as extensive editing required.

Major points

1. HeLa cells were used for all functional experiments, but RPE1 cells were used for immunofluorescence experiments in some cases. This requires explanation. Whether similar results were obtained in both cell types should be stated.
2. Figure 3E lacks a crucial rescue control with sh-ADD1/S726D or sh-ADD1/WT to support that ADD1 plays a role in centriole adhesion. The statistical significance of distribution differences should be tested (e.g. by chi-square).
3. Figure 4A, "The S726 phosphorylation of ADD1 was measured and expressed as the fold increase relative to the level in the asynchronous HeLa cells." This normalization cannot account for ADD1 level variation among samples. pS726 ADD1 should be normalized against the total ADD1 level in each sample, and then compared. If the authors think the input levels of ADD1 cross all samples are similar, quantified blots should be provided in a supplemental figure.
4. Figure 4B does not support that Plk1 phosphorylate ADD1 S726. Plk1 inhibition suppresses centrosome maturation, as indicated by the weak gamma-tubulin signal. Therefore, the reduced ADD1 pS726 signal could be caused by a decrease in PCM recruitment and not phosphorylation. The signal should be normalized to the total ADD1 signal at centrosomes.
5. All Co-IP quantifications should be provided in a supplemental figure.
6. Figure 7C lacks the important quantification of control and sh-ADD1/WT as well as proper statistical analysis. Figure 7 is barely described in the main text and requires more explanation.
7. S6A is not convincing given the high background. A negative control protein that is localized on the spindle but not at centrosomes should be included. Most importantly, S6B needs a negative control.

8. Page 11, 1st paragraph line 8, "In this study, we demonstrate that..." After submission, a study was published showing that TPX2 is also centrosomal (DOI: 10.1016/j.cub.2017.11.046, PMID: 29276125). The authors should now cite this paper.

9. Page 12, 1st paragraph: the authors should consider discussing the results from Jones et al, 2014 in this context, since silencing Dynein, which is counteracted by Eg5, has very similar phenotype to that reported in this manuscript.

Additional presentation issues (incomplete list). Line numbering would have been very helpful!

1. Page 3: PCMs is not used (just PCM)
2. Page 6, 1st paragraph: more explanation required - the pS726 peptide is a competitor for the phosphospecific antibody.
3. Page 6: "...localized to centrosomes more apparently..." not standard
4. Page 6, 2nd paragraph 3rd line: "In G2-phase centrosomes," is not consistent with the figure legend which says "interphase".
5. Page 7: The Aurora B depletion control requires more explanation.
6. Page 8, 2nd paragraph 2nd line: "200 nM" is inconsistent with Figure 4D.
7. Page 9, 1st paragraph 3rd line: The statement that "TPX2 interacted with ADD1 during mitosis but not during interphase (Fig. 5A)" is too strong, as the TPX2 level is very low during interphase. Could be: "The interaction between ADD1 and TPX2 could not be detected during interphase..."
8. Figure S6: centrosomal is misspelled in the figure.

Referee #3:

The current manuscript describes the phosphorylation of Adducin 1 on Serin 726 and its molecular function at the centrosome. The authors use knock-down - rescue approaches, pulldowns and biochemical experiments combined with high resolution microscopy as well as phenotypic analysis of spindle morphology/integrity to analyse the significance of Adducin S726 phosphorylation in mitotic cells. Using phospho-specific antibodies, they show that S726-phosphorylated Adducin accumulates at centrosomes in a mitosis-specific manner. wt Adducin rescues defects in spindle pole integrity after knockdown of endogenous Adducin (previously described by Chan et al, JCB, 2014) but not the S726A mutant. Further evidence supports the idea that Plk1 phosphorylates Adducin1 on S726. To show the mechanism how S726 phosphorylation might act on spindle pole integrity, the authors identify the interaction of Adducin 1 with TPX2. They show that variants of TPX2, unable to interact with Adducin 1, cannot restore spindle pole integrity after TPX2 knockdown.

Taken together, this study is interesting and provides significantly novel information about the function of Adducin 1 in mitosis. All conclusions are very solid; they rely on experiments of extraordinary technical quality with careful controls and numerous quantifications. Technical excellence, in particular, argues for publication of the manuscript in EMBO R.

One major experimental/conceptual issue still remains to be answered. The authors draw an image in which TPX2 and Adducin 1 interact with each other at spindle poles/centrosomes. The fact that the interaction is stimulated by the S726 phosphorylation (only found at centrosomes) suggests that they do interact here, however, this is a limited view, due to my opinion, and could be modified as follows:

It should be more clearly stated that the S726A mutant preserves the interaction with TPX2. How does the S726D mutant look like in this assay? Don't the data indicate an interaction of Adducin with TPX2 at least also along spindle microtubules? The localisation of tagged Adducin wt vs. the S726 mutants (suppl info) would be very valuable but is not very informative yet; possibly other tags work better for IF to show these localisations?

It may be misleading to restrict the description of colocalization to images of cells in which spindle microtubules are depolymerised by cold shock (see Fig. 5D). It may help to show not only colocalization of P-S726 and TPX2 at spindle poles but also of the bulk of Adducin with TPX2 along spindle microtubules. The function in spindle pole integrity of TPX2 also relies on its role in microtubule nucleation on spindle microtubules (Bird et al, recent publication of Zhang et al, ELife,

2017). The dynamic interaction with Adducin on spindle poles AND spindle microtubules might therefore be the desired take home message.

While the interaction domain in TPX2 for Adducin 1 interaction was determined, we do not exactly know which elements in Adducin 1 promote binding to TPX2. Stimulation by S726 phosphorylation may suggest that the interaction requires the C-terminus of Adducin 1 (?). A domain analysis should be shown.

Additional concerns:

The fact that TPX2 recruits to spindle poles and is required for spindle pole integrity is certainly not a new observation but has been demonstrated in several publications in the past (Wittmann et al., Garrett et al., Bird et al., etc.).

Fig. 4B: why are the g-tub signals way stronger in monastrol conditions then under BI2536?

Fig. 3E and 7C show only some of the conditions. While it may not be necessary to quantify all conditions, Fig. 3E should include the control and S727D and Fig. 7C the wt rescue experiment.

The individual paragraphs of the ms are often not well connected, e.g. the introduction reads a bit like a recital of mitotic factors; this should be better connected. The same in the results: the analysis of TPX2 as a potential interaction partner is hardly motivated.

The statistics part mentions a students T-test to evaluate significance, but does not explain any further detail (paired, unpaired etc.) neither gives sample sizes, which are also not mentioned in figure legends.

The methods section is very detailed, which is desirable, but may still be cut to become more concise.

1st Revision - authors' response

17 April 2018

Referee #1:

Major points:

1. The centriole splitting phenotype shown in Fig. 3 could arise from the extended mitotic delay previously reported in ADD1-depleted cells by Chan et al., 2014 JCB. The authors should perform time-lapse microscopy on cells expressing fluorescent reporters that allow visualization of spindle morphology. This would reveal if multipolarity occurs upon entry into mitosis or only later during the mitotic arrest. They could compare/contrast these to Myo10- and TPX2-depleted cells. Also, the authors suggest that the role of TPX2 in spindle pole integrity has not been elucidated, but this is not the case and they should cite papers that described centriole splitting in TPX2-depleted cells previously (i.e. de Luca et al., Cell cycle, 2006). The phenomenon was already described in Garrett et al., 2002, which the authors cite but only for describing its role in spindle bipolarity.

Response:

- (1) Thank the reviewer for raising this critical issue. We performed the time-lapse microscopy and found that 50.9% and 36.8% of the multipolar spindles caused by ADD1 and TPX2 depletion occurred within one hour upon entry into mitosis, respectively (Figure EV4). In contrast, Myo10 depletion caused spindle distortion rather than multipolar spindles (Figure EV4). Therefore, the multipolar spindles caused by ADD1 cannot totally attribute to a mitotic arrest
- (2) We cited the references and revised the manuscript accordingly.

2. Although the authors show that ADD1 and TPX2 co-sediment in sucrose fractionation, it would be important to demonstrate that endogenous ADD1 co-immunoprecipitates with endogenous TPX2. Currently, one partner is always overexpressed and there is no in vitro data to support direct interaction between the two proteins. If antibodies are not suitable for immunoprecipitation, a tag could be knocked into the ADD1 locus.

Response:

We have tried to detect the co-immunoprecipitation of endogenous ADD1 and TPX2, but not successful. Overexpression of FLAG-ADD1 is necessary to co-precipitate endogenous TPX2. We show that GST-TPX2 interacts with purified FLAG-ADD1 in vitro (new Figure 4F), supporting a direct interaction between TPX2 and ADD1.

3. From the manuscript it is unclear why the authors tested whether ADD1 could bind TPX2. There should be a clearer explanation in the text, but for now I can only assume it is because TPX2 was previously shown to bind Myo10, and Myo10 is also a binding partner of ADD1. If this is so, it is surprising that there is no attempt in the paper to investigate whether these proteins form a ternary complex. Does Ser726 phosphorylation interfere with binding of ADD1 to Myo10? How do Ser12 and Ser355 phosphorylations affect TPX2 binding? What is the hierarchy of centrosome and spindle recruitment by Myo10/TPX2/ADD1? Does ADD1 also co-immunoprecipitate with Aurora-A? Does ADD1 overexpression/depletion affect Aurora-A binding to the mitotic spindle (i.e. by sequestration of TPX2)? Addressing these questions would provide mechanistic insight and improve the novelty and impact of the study.

Response:

- (1) Given that Myo10 interacts with TPX2 (Woolner et al., 2008) and ADD1 (Chan et al., 2014), this prompted us to examine whether TPX2 interacts with ADD1 through Myo10. However, we cannot demonstrate this ternary complex; instead, our results support a direct interaction between ADD1 and TPX2. In addition, mutating S726 does not affect ADD1 interaction with Myo10 (Appendix Figure S5). Likewise, ADD1 with mutations at both S12 and S355 retained its interaction with TPX2 (Appendix Figure S5). These data suggest that different subsets of ADD1 may preferentially interact with TPX2 and Myo10 through phosphorylation at different serine residues by different mitotic kinases. In this scenario, the ADD1^{pS726} interacts with TPX2 at mitotic centrosomes for the spindle pole integrity, whereas ADD1^{pS12/S355} interacts with Myo10 at spindle fibers for proper spindle assembly.
- (2) As suggested by the reviewer, we examined the interaction of ADD1 with Aurora-A. We found that ADD1 indeed interacts with Aurora-A in mitosis, but not in interphase (data not shown). Since we still do not know the significance of this interaction, I do not think it is appropriate to include this piece of information into this manuscript. More efforts are certainly needed to clarify the interactions between ADD1, Aurora-A, and TPX2 and the functional significance.
- (3) As suggested by the reviewer, we examined the effect of ADD1 depletion on the spindle localization of Aurora-A. We found that Aurora-A retained its association with bipolar and multipolar spindles after ADD1 depletion (data not shown).

4. Related to point 3, the authors need to place their finding that TPX2 and ADD1 are in a protein complex into a broader picture, as it is hardly surprising that two proteins with a common binding partner might interact. The model in Fig. 8 should be modified accordingly.

Response:

We modified our model in new Figure 7F.

5. The authors suggest that Ser726 of ADD1 is phosphorylated by PLK1. Based on Fig. 4A the BI2536 treatment only slightly reduced levels of phospho-T210 PLK1, which may explain why there is only a small reduction in phospho-Ser726. One would expect a more complete response to this highly specific inhibitor. Centrosomal levels of ADD1 are clearly affected by BI2536 treatment, but PLK1 inhibition is known to disrupt the PCM, so this may not be a direct effect. Phosphorylations of ADD1 could also be mapped by MS in absence/presence of BI2536. These are not essential experiments for the paper as a whole, but if not performed, it might be wiser to remove this figure.

Response:

We thank the reviewer for raising this important issue. We performed centrosomal fractionation experiments and found that the decrease in the ADD1 pS726 by BI2536 treatment is likely to be the result from a decreased level of ADD1 being recruited to mitotic centrosomes upon PLK1 inhibition (data not shown). Besides, overexpression of PLK1 did not increase ADD1 S726 phosphorylation (data not shown). Therefore, PLK1 is likely not the kinase responsible for ADD1 S726 phosphorylation during mitosis. As suggested by the reviewer, we remove the old Figure 4.

6. The term 'spindle pole integrity' is used in different contexts in the text. For instance, in the last paragraph the authors use it to describe shorter spindles. These should be clarified.

Response:

The loss of 'spindle pole integrity' is usually used to describe two types of mitotic defects; PCM fragmentation and centriole splitting. In this study, the term 'spindle pole integrity' only refers the maintenance of the spindle pole with a pair of centrioles. We proofread the manuscript to avoid the confusion.

7. Statistical tests are needed for all experimental groups in Fig. 6 E and F. Based on Fig. 6E binding by TPX2 to Eg5 or ADD1 is redundant in terms of spindle length maintenance as either deletion construct is capable of rescuing spindle length in shTPX2-treated cells, so the authors should not use the term "essential". Since interaction of TPX2 with CLASP is also implicated in spindle length regulation, I wonder if the authors excluded the possibility that CLASP and ADD1 interact the same domain of TPX2.

Response:

- (1) As suggested by the reviewer, we used the term "important" to replace the term "essential" for description of the results.
- (2) Thank the reviewer for pointing out CLASP1 as a TPX2 binding partner. Phosphorylation of TPX2 at Ser121 and Ser125 by Aurora A is important for its interaction with CLASP1 (Fu et al., 2015), but the region of TPX2 for CLASP1 binding remains unclear. In this study, we found that the aa 120-370 of TPX2 is sufficient for its interaction with ADD1. Due to the proximity of Ser121 and Ser125 to the ADD1-binding region, it is possible that the CLASP1-binding region may be proximal or overlapped with the ADD1-binding region. Thus, the possibility that ADD1 and CLASP1 mutually affect their binding to TPX2 cannot be excluded. More experiments are needed to clarify this issue. However, the TPX2 mutant (with a deletion of aa. 236-370) deficient in ADD1 binding that retains Ser121 and Ser125 fails to rescue the defect of multipolar spindle caused by TPX2 depletion (new Figure 5), suggesting that CLASP1 binding, if there is any, is not sufficient to restore the defect. In fact, the TPX2-CLASP1 interaction is known to be important for the control of spindle length (Fu et al., 2015). Therefore, our results still support the conclusion that the TPX2-ADD1 interaction is important for the spindle pole integrity.

Fu, J., M. Bian, G. Xin, Z. Deng, J. Luo, X. Guo, H. Chen, Y. Wang, Q. Jiang, and C. Zhang. 2015. TPX2 phosphorylation maintains metaphase spindle length by regulating microtubule flux. *J. Cell Biol.* 210:373-383.

Minor points:

1. Labelling in figures should be more complete (e.g. Fig5E: include which protein/fragment is tagged with what).

Response:

All figure labeling is checked and modified, if necessary.

2. Fig. 4A: second blot, correct IB: Auroa A to Aurora A.

Response:

The typo is corrected.

Referee #2:

1. HeLa cells were used for all functional experiments, but RPE1 cells were used for immunofluorescence experiments in some cases. This requires explanation. Whether similar results were obtained in both cell types should be stated.

Response:

We also performed the immunofluorescence staining in HeLa cells and obtained similar results as in RPE1 cells (new Appendix Figure S2). The reason to show the immunofluorescence images of RPE1 cells in the main Figure 1 is because the image background is lower in RPE1 cells than in HeLa cells.

2. Figure 3E lacks a crucial rescue control with sh-ADD1/S726D or sh-ADD1/WT to support that ADD1 plays a role in centriole adhesion. The statistical significance of distribution differences should be tested (e.g. by chi-square).

Response:

- (1) In Figure 3E, the number of centrioles was only measured in the pole of multipolar spindles. Because ADD1 WT and S726D are able to rescue the defect of multipolar spindles and the spindles in the control cells, sh-ADD1/S726D, and sh-ADD1/WT cells are bipolar, those cells are not included in the assessment for Figure 3E.
- (2) The statistical significance of distribution differences was determined by unpaired Student's t-test.

3. Figure 4A, "The S726 phosphorylation of ADD1 was measured and expressed as the fold increase relative to the level in the asynchronousized HeLa cells." This normalization cannot account for ADD1 level variation among samples. pS726 ADD1 should be normalized against the total ADD1 level in each sample, and then compared. If the authors think the input levels of ADD1 cross all samples are similar, quantified blots should be provided in a supplemental figure.

Response:

As suggested by the reviewer, the ADD1 pS726 is normalized to the ADD1 level.

4. Figure 4B does not support that Plk1 phosphorylate ADD1 S726. Plk1 inhibition suppresses centrosome maturation, as indicated by the weak gamma-tubulin signal. Therefore, the reduced ADD1 pS726 signal could be caused by a decrease in PCM recruitment and not phosphorylation. The signal should be normalized to the total ADD1 signal at centrosomes.

Response:

Thank the reviewer for raising this important issue. We performed centrosomal fractionation experiments and found that the decrease in the ADD1 pS726 by BI2536 treatment is likely to be the result from a decreased level of ADD1 being recruited to mitotic centrosomes upon PLK1 inhibition (data not shown). Besides, overexpression of PLK1 did not increase ADD1 S726 phosphorylation (data not shown). Therefore, PLK1 is likely not the kinase responsible for ADD1 S726 phosphorylation during mitosis. As suggested by the reviewer #1, we remove the old Figure 4.

5. All Co-IP quantifications should be provided in a supplemental figure.

Response:

All Co-IP quantifications are now available in Source data.

6. Figure 7C lacks the important quantification of control and sh-ADD1/WT as well as proper statistical analysis. Figure 7 is barely described in the main text and requires more explanation.

Response:

As in Figure 3E, the number of centrioles was only measured in the pole of multipolar spindles. Because the control cells and sh-ADD1/WT cells display bipolar spindles, they are not included in the assessment for Figure 7C (new Figure 6C). Depletion of Aurora-B was used as the control. The statistical significance was determined and indicated in the Figure. As suggested, we describe this figure in more detail in the revised manuscript.

7. S6A is not convincing given the high background. A negative control protein that is localized on the spindle but not at centrosomes should be included. Most importantly, S6B needs a negative control.

Response:

The immunofluorescence images in old Fig. S6A were replaced by new ones (Appendix Figure S4A). We performed new centrosome fractionation experiments and included a negative control to the experiment (Appendix Figure S4B).

8. Page 11, 1st paragraph line 8, "In this study, we demonstrate that..." After submission, a study was published showing that TPX2 is also centrosomal (DOI: 10.1016/j.cub.2017.11.046, PMID: 29276125). The authors should now cite this paper.

Response:

As suggested by the reviewer, the reference is cited in the revised manuscript.

9. Page 12, 1st paragraph: the authors should consider discussing the results from Jones et al, 2014 in this context, since silencing Dynein, which is counteracted by Eg5, has very similar phenotype to that reported in this manuscript.

Response:

Thank the reviewer for the suggestion. We cite the reference and discuss this issue in the Discussion section.

Minor points:

1. Page 3: PCMs is not used (just PCM)

Response:

The Typo was corrected.

2. Page 6, 1st paragraph: more explanation required - the pS726 peptide is a competitor for the phosphospecific antibody.

Response:

As suggested by the reviewer, we added this information to the text.

3. Page 6: "...localized to centrosomes more apparently..." not standard

Response:

As suggested, we modified the sentence in the revised manuscript.

4. Page 6, 2nd paragraph 3rd line: "In G2-phase centrosomes," is not consistent with the figure legend which says "interphase".

Response:

As suggested, we changed the term "G2-phase" to "interphase" in the revised manuscript.

5. Page 7: The Aurora B depletion control requires more explanation.

Response:

As suggested, we explain more about why we used the depletion of Aurora-B as the control in the revised manuscript.

6. Page 8, 2nd paragraph 2nd line: "200 nM" is inconsistent with Figure 4D.

Response:

The mistake was corrected.

7. Page 9, 1st paragraph 3rd line: The statement that "TPX2 interacted with ADD1 during mitosis but not during interphase (Fig. 5A)" is too strong, as the TPX2 level is very low during interphase. Could be: "The interaction between ADD1 and TPX2 could not be detected during interphase..."

Response:

As suggested by the reviewer, the sentence was modified.

8. Figure S6: centrosomal is misspelled in the figure.

Response:

The typo was corrected.

Referee #3:

1. One major experimental/conceptual issue still remains to be answered. The authors draw an image in which TPX2 and Adducin 1 interact with each other at spindle poles/centrosomes. The fact that the interaction is stimulated by the S726 phosphorylation (only found at centrosomes) suggests that they do interact here, however, this is a limited view, due to my opinion, and could be modified as follows: It should be more clearly stated that the S726A mutant preserves the interaction with TPX2. How does the S726D mutant look like in this assay? Don't the data indicate an interaction of Adducin with TPX2 at least also along spindle microtubules? The localisation of tagged Adducin wt vs. the S726 mutants (suppl info) would be very valuable but is not very informative yet; possibly other tags work better for IF to show these localisations?

Response:

The immunofluorescence images in old Fig. S6A were replaced by new ones (Appendix Figure S4A). Like its wild-type counterpart, the FLAG-ADD1 S726A mutant localizes to mitotic centrosomes, as analyzed by immunofluorescence staining (Appendix Figure S4A) and centrosome fractionation (Appendix Figure S4B), indicating that the phosphorylation of ADD1 at S726 is not required for its localization to mitotic centrosomes. These data also suggest that ADD1 may be first recruited to mitotic centrosomes via a not-yet-known mechanism and then phosphorylated at S726 by a protein serine kinase, which thereby promotes the interaction of ADD1 with TPX2 in these

subcellular compartments. Although ADD1 and TPX2 localize at both spindle poles and fibers, we have no evidence to demonstrate their interaction along the spindle fibers.

2. It may be misleading to restrict the description of colocalization to images of cells in which spindle microtubules are depolymerised by cold shock (see Fig. 5D). It may help to show not only colocalization of P-S726 and TPX2 at spindle poles but also of the bulk of Adducin with TPX2 along spindle microtubules. The function in spindle pole integrity of TPX2 also relies on its role in microtubule nucleation on spindle microtubules (Bird et al, recent publication of Zhang et al, ELife, 2017). The dynamic interaction with Adducin on spindle poles AND spindle microtubules might therefore be the desired take home message.

Response:

We show that mutating S726 does not affect ADD1 interaction with Myo10 (Appendix Figure S5). Likewise, ADD1 with mutations at both S12 and S355 retained its interaction with TPX2 (Appendix Figure S5). These data suggest that different subsets of ADD1 may preferentially interact with TPX2 and Myo10 through phosphorylation at different serine residues by different mitotic kinases. In this scenario, the ADD1pS726 interacts with TPX2 at mitotic centrosomes for the spindle pole integrity, whereas ADD1pS12/S355 interacts with Myo10 at spindle fibers for proper spindle assembly.

3. While the interaction domain in TPX2 for Adducin 1 interaction was determined, we do not exactly know which elements in Adducin 1 promote binding to TPX2. Stimulation by S726 phosphorylation may suggest that the interaction requires the C-terminus of Adducin 1. A domain analysis should be shown.

Response:

As suggested by the reviewer, we performed new experiments and found that GST-TPX2 aa 120~370 is sufficient to bind purified FLAG-ADD1 in vitro (new Figure 4F), supporting a direct interaction between TPX2 and ADD1. In addition, we demonstrated that GST-TPX2 aa 120~370 binds the tail domain of ADD1, but not the ADD1 mutant with a deletion of the tail domain (new Figure 4G). Therefore, the tail domain of ADD1 is the region for TPX2 binding.

Additional concerns:

1. The fact that TPX2 recruits to spindle poles and is required for spindle pole integrity is certainly not a new observation but has been demonstrated in several publications in the past (Wittmann et al., Garrett et al., Bird et al., etc.).

Response:

As suggested by the reviewer, the references are cited in the revised manuscript.

2. Fig. 4B: why are the γ -tub signals stronger in monastrol conditions than under BI2536?

Response:

PLK1 inhibition is known to suppress centrosome maturation, which is likely the reason why the γ -tubulin signal is much lower in BI2536-treated centrosomes than in the monastrol-treated ones.

3. Fig. 3E and 7C show only some of the conditions. While it may not be necessary to quantify all conditions, Fig. 3E should include the control and S727D and Fig. 7C the wt rescue experiment.

Response:

In Figure 3E and 7C (new Figure 6C), the number of centrioles was only measured in the poles of multipolar spindles. Because ADD1 WT and S726D are able to rescue the defect of multipolar spindles and thereby the spindles in the control cells, sh-ADD1/S726D, and sh-ADD1/WT cells are mainly bipolar, those cells are not included in the assessment for both figures.

4. The individual paragraphs of the ms are often not well connected, e.g. the introduction reads a bit like a recital of mitotic factors; this should be better connected. The same in the results: the analysis of TPX2 as a potential interaction partner is hardly motivated.

Response:

- (1) We revised our manuscript to make it more connected between paragraphs.
- (2) Given that Myo10 interacts with TPX2 (Woolner et al., 2008) and ADD1 (Chan et al., 2014), this prompted us to examine whether TPX2 interacts with ADD1 through Myo10. However, we cannot demonstrate this ternary complex; instead, our results support a direct interaction between ADD1 and TPX2.

5. The statistics part mentions a student's T-test to evaluate significance, but does not explain any further detail (paired, unpaired etc.) neither gives sample sizes, which are also not mentioned in figure legends.

Response:

The detailed information for quantitative results is available in the revised figure legends.

6. The methods section is very detailed, which is desirable, but may still be cut to become more concise.

Response:

As suggested by the reviewer, we modified the methods section to make it more concise. Some detailed information is described in Appendix Supplementary Methods.

2nd Editorial Decision

18 May 2018

Thank you for the submission of your revised manuscript to EMBO reports. We have now received the full set of referee reports that is copied below.

As you will see, all three referees are positive about the study and support publication in EMBO reports after some minor textual changes. Referee 1 suggests including the data on the interaction between ADD1 and Aurora-A and I leave this decision to you.

Browsing through the manuscript myself, I noticed a few things that we need before we can proceed with the official acceptance of your manuscript:

- I noticed that the Appendix contains supplementary methods. Please note that all material and methods have to be part of the main manuscript text, unless they are of very specialized interest. I don't think that this is the case here and I therefore kindly ask you to incorporate them into the main manuscript. If you wish, the description of subcloning the S726D mutant or of TPX2 into the different vectors could remain in the Appendix but preferentially, it is also moved to the main text.

- Figure Callouts: Please note that all figures and figure panels should be arranged in the order in which they are mentioned in the text. I noticed that Fig. 5C is described in the text before 5B. Also Fig 7D + E are called-out before 7B + C and I therefore suggest to swap these panels in the figure. Moreover, the different panels shown in Fig 6 are not called out at all in the text.

- Was the mass spectrometry data generated for this study? If so, please describe the experimental details in the Materials and Methods section - they appear to be missing. If the data were taken from an earlier study, please cite this paper.

- Our data editors have already checked the figure legends for completeness and clarity and made some suggestions on how to improve it (see attached file). Moreover, our routine text analysis indicated two sentences that are very similar to a previously published article and I suggest to rephrase this part.

- Finally, EMBO reports papers are accompanied online by A) a short (1-2 sentences) summary of the findings and their significance, B) 2-3 bullet points highlighting key results and C) a synopsis image that is 550x200-400 pixels large (width x height). You can either show a model or key data in the synopsis image. Please note that the size is rather small and that text needs to be readable at the final size. Please send us this information along with the revised manuscript.

REFEREE REPORTS

Referee #1:

The authors have addressed my criticism and improved the manuscript accordingly. It is disappointing that they were unable to demonstrate co-immunoprecipitation of endogenous ADD1 and Tpx2, but the in vitro binding data seems supportive of a direct interaction.

Minor points:

1. The authors appear cautious about their results regarding the ADD1 and Aurora-A interaction (described in the rebuttal), but now that the Plk1 data has been removed, they may want to consider including these.
2. Although the sentence that the interaction of Tpx2 with ADD1 and Eg5 is 'essential' for spindle integrity has indeed been changed to 'important' in the text, the title of this section still states 'The interaction of TPX2 with both ADD1 and Eg5 is ESSENTIAL for spindle pole integrity'.

Referee #2:

The authors have addressed my concerns and the manuscript is now suitable for publication in EMBO Reports.

Referee #3:

In their revised manuscript, Hsu et al. addressed major concerns that I had raised in my initial report adequately. In particular, I appreciate the experimental effort to show that the C-terminus of Adducin interacts with TPX2. Further analysis of the interaction of TPX2 and Adducin along spindle microtubules, and finding out more about the significance of the latter, would have been desirable but I do concede that this goes beyond the scope of this study. Taken together, and also judging the responses to all reviewers' criticism, I strongly suggest publication of this insightful study in EMBO Reports.

2nd Revision - authors' response

21 May 2018

I have modified the manuscript according to your instructions provided in the Decision Letter. I hope you will feel that the revisions are appropriate for publication.

Corresponding Author Name: Hong-Chen Chen

Manuscript Number: EMBOR-2017-45607-T